# Modelling the late-Holocene and future evolution of Monacobreen, northern Spitsbergen

Johannes Oerlemans[1]

[1]Institute for Marine and Atmospheric Research, Utrecht University, Princetonplein 5, Utrecht, 3585CC, The Netherlands

*Correspondence to*: Johannes Oerlemans (j.oerlemans@uu.nl)

## Abstract

Monacobreen is a 40 km long surge-type tidewater glacier in northern Spitsbergen. During 1991-1997 Monacobreen surged and advanced by about 2 km, but the front did not reach the maximum Little Ice Age (LIA) stand. Since 1997 the glacier front is retreating at a fast rate (~ 125 m/a). The questions addressed in this study are: (1) Can the late Holocene behaviour of Monacobreen be understood in terms of climatic forcing ?, and (2) What will be the likely evolution of this glacier for different scenarios of future climate change ?

Monacobreen is modelled with a Minimal Glacier Model, including a parameterization of the calving process as well as the effect of surges. The model is driven by an Equilibrium Line Altitude (ELA) history derived from lake sediments of a nearby glacier catchment, in combination with meteorological data from 1899 onwards. The simulated glacier length is in good agreement with the observations: the maximum LIA stand, the front position at the end of the surge, and the 2.5 km retreat after the surge (1997-2016) are well reproduced. The effect of surging is limited. Directly after a surge the initiated mass-balance pertubation due to a lower mean surface elevation is about $-0.13$ m w.e. a$^{-1}$, which only has a small effect on the long-term evolution of the glacier. The simulation suggests that the major growth of Monacobreen after the Holocene Climatic Optimum started around 1500 BCE. Monacobreen became a tidewater glacier around 500 BCE, and reached a size comparable to the present state around 500 CE. For the mid-B2 scenario (IPCC, 2013), which corresponds to a ~2 m a$^{-1}$ rise of the ELA, the model predicts a volume loss of 20 to 30 % by the year 2100 (relative to the 2017 volume). For a ~4 m a$^{-1}$ rise in the ELA this is 30 to 40 %. However, much of the response to 21st century warming will still come after 2100.

## 1 Introduction

In view of the observed strong warming of the polar regions, the future evolution of arctic ice masses is of great concern. Glacier retreat has a huge impact on regional ecosystems, and it contributes significantly to sea-level rise. The large ice sheets of Greenland and Antarctica are losing mass (IPCC, 2013; Van den Broeke et al., 2016; Martin-Español et al., 2017), and glaciers have retreated far behind their Little Ice Age (LIA) maximum stands almost everywhere (Leclercq et al. 2014; Zemp et al., 2015). Even the large tidewater glaciers of the Arctic region have retreated over large distances (kilometers) during the past 100 years. Although tidewater and surging glaciers have a strong internal component to their dynamics, it has become clear that climatic forcing will now determine to a large extent their future evolution.

Different approaches have been taken to model the future behaviour of glaciers. Individual glaciers have been studied with flowline models based on approximate formulations of the mechanics of ice flow (e.g. Kruss, 1984; Huybrechts et al., 1989; Oerlemans, 1997; Nick et al., 2007). In more recent years, three-dimensional higher-order models have been employed, for

instance for the Rhone glacier (Jouvet et al., 2009), and for the Morteratsch glacier (Zekkolari et al., 2013). Smaller ice caps are also being studied with such models (e.g. Zekkolari et al., 2017; Schäfer et al. (2015).

The projection of future glacier changes depends strongly on the definition of the initial state used to start the integration.
Glaciers have response times of decades to centuries, implying that a reliable projection of future evolution can only be achieved when changes over a past timespan can first of all be reproduced (termed "dynamic calibration" in Oerlemans, 1997). This timespan should have a duration of at least the response time, preferably longer. In Oerlemans et al. (1998) this approach was applied to a sample of 12 glaciers, and a rather simple attempt was made to apply the results to the entire glacier population on the globe.

Along a different line, models have been constructed to treat the entire glacier population on the globe (e.g. Radic et al., 2014; Huss and Hock 2015). In these models the focus is on estimating global glacier mass changes as a response to warming and its effect on sea-level rise. Processes like accumulation, melting, calving and dynamic adjustments of glacier geometries are treated in a schematic way. The models have a large number of empirical parameters, and are tuned by comparison with observed mass balance data over the past decades. Clearly, when questions of a global nature have to be
dealt with, global models are needed in which not every single glacier can be treated in detail. Nevertheless, it seems that in studies of this kind historical data on glacier fluctuations over the past centuries have not yet been fully exploited.

The modelling strategies discussed above have their advantages as well as their limitations. The three-dimensional models require a large amount of geometric input data, and the issue of boundary conditions in steep terrain has not yet been resolved in a satisfactory way (dynamic simulation of steep ice-free slopes, mass conservation). There appears to be room
for an intermediate approach with simpler models for individual glaciers that still take into account a number of essential feedback mechanisms (altitude-mass balance feedback, effect of overdeepenings, variable calving rates, effect of regular surging on the long-term mass budget, inclusion of tributary glaciers and basins).

Minimal Glacier Models (MGM's, Oerlemans 2011) offer the possibility to model individual glaciers in a relatively simple way, while still dealing with mechanisms like altitude - mass balance feedback, the effect of overdeepenings in the bed,
variable calving rates, the effect of regular surging on the long-term mass budget, etc. MGMs require limited computational resources and can therefore be used efficiently in control methods where many integrations have to be carried out. MGM's have been applied to MacGall glacier in Alaska, (Oerlemans, 2011), to Hansbreen, a calving glacier in southern Spitsbergen (Oerlemans et al., 2011), and to Abrahamsenbreen, a surging glacier in northern Spitsbergen (Oerlemans and Van Pelt, 2015).

In this paper, a MGM is applied to Monacobreen, a 40 km long glacier in northern Spitsbergen that flows from the plateau Isachsenfonna (~1000 m a.s.l.) into the Liefdefjord (glacier coordinates 79°23' N, 12°38' E; see the interactive topographic / thematic map operated by the Norsk Polarinstitutt: http://toposvalbard.npolar.no). The glacier drains an area of about 400 km$^2$, and the main stream has an average slope of only 0.027. Modelling Monacobreen is difficult, because it is a surge-type calving glacier with many tributary glaciers / basins. Some information is available on former positions of the glacier front
(Fig. 1). From 1991 till 1997 the glacier surged, leading to a 2 km advance of the glacier front (Mansell et al., 2012). Røthe et al. (2015) made a reconstruction of the Equilibrium Line Altitude (ELA) for the Holocene, derived from a thorough analysis of lake sediments in the catchment of the small glacier Karlbreen in northwest Spitsbergen. Karlbreen is located only about 25 km to the west of Monacobreen, and the ELA reconstruction thus offers a unique possibility to simulate the evolution of Monacobreen through Holocene times.

Monacobreen, with a distinct calving main stream and a large number of tributaries, represents a glacier system that is typical for Spitsbergen. The complexity of the geometry as well as the limited amount of data make it a real challenge for a modelling study. Nevertheless, the question of how the mass of such glacier systems will change in the near future has to be

considered, and the approach taken in this study is an attempt to do this in a meaningful way. A MGM provides a reasonable match between the paucity of data and an integrated mass budget approach, in which glacier mechanics are parameterized in a simple way. The larger glacier systems on Svalbard presumably have long response times, because they have small slopes and are subject to relatively dry conditions. The strategy of using observations on former glacier stands for calibration before integrating the model into the future is tested in this paper. It is envisaged that the methodology can be applied to other complex glacier systems in Svalbard and the Arctic.

With respect to Monacobreen, the following more specific questions will be addressed: (i) Is it possible to simulate the broad characteristics of the late Holocene evolution of Monacobreen ?; (ii) To what extent does regular surging effect the mass budget and long-term evolution of the glacier ? ; and (iii) What is the likely range of mass loss in the coming centuries for different scenarios of climate change ?

## 2 Glacier model

Monacobreen is considered to be a stream (flowband) of length $L$ and constant width $W$, to which 10 tributary glaciers / basins supply mass if the equilibrium line is sufficiently low (Fig. 2). The length is measured along the $x$-axis, which follows the centerline of the flowband. With a width of 5 km and a length of 40 km, the area of the main stream is about 200 km². Currently the area of the tributary basins is 191 km² (not including Seligerbreeen), and since these basins have a higher mean suface elevation than the main stream, they make a major contribution to the total mass budget. The number of basins that actually feed the main stream depends on $L$.

In the following sections a number of parameterizations are introduced concerning the global ice mechanics, geometry, calving, and climate forcing. An overview of the parameters and their values (including the sources) is given in Table 1.

### 2.1 Basic formulation

The evolution of the glacier system is described by the conservation of mass (or volume, since ice density is considered to be constant). Since a distinction is made between the main stream and tributary glaciers, the governing equation is conveniently written as:

$$\frac{dV}{dt} = F + B_M + \sum_{i=1}^{10} B_i = B_{tot} \ ,$$

(1)

where $V$ is the volume of the main stream of Monacobreen, $F$ is the calving flux ($< 0$), $B_M$ is the surface mass budget of the main stream, and the $B_i$ are the contributions from the tributary glaciers. The total mass budget of the main stream is denoted by $B_{tot}$. The calculation of the mass fluxes from the tributaries is discussed in section 2.2.

The glacier volume $V$ (of the main stream) is given by $WLH_m$ , where $H_m$ is the mean ice thickness. Differentiating with respect to time yields

$$\frac{dV}{dt} = W \frac{d}{dt}(LH_m) = W \left( H_m \frac{dL}{dt} + L \frac{dH_m}{dt} \right) = B_{tot} \ .$$

(2)

The mean ice thickness is written as (Oerlemans, 2011)

$$H_m = S(t) \frac{\alpha}{1 + v\bar{s}(L)} L^{1/2} .$$

(3)

Here $\bar{s}$ is the mean bed slope over the glacier length, and thus varies in time. $S$ is the "surge function", making it possible to impose a surge cycle. A rapid decrease in $S$ leads to a reduction of the mean ice thickness and consequently an increase in the glacier length to fulfill mass conservation. This technique was succesfully applied in the study of Abrahamsenbreen

(Oerlemans et al., 2011). Eq. (3) is based on extensive numerical experimentation with a Shallow Ice Approximation model, including a Weertman type sliding law. For many glaciers the parameter values $\nu = 10$ and $\alpha = 3$ m$^{1/2}$ work well, but the larger glaciers in northern Svalbard flow over beds with relatively low resistence and smaller values of $\alpha$ apply (to be discussed later). Note that in the limit for $\bar{s} \longrightarrow 0$ the mean thickness varies with the square root of the glacier length, which is in agreement with the standard analytical theory for plane shearing flow (Vialov, 1958).

After some straightforward algebraic manipulation (Oerlemans, 2011), the prognostic equation for glacier length can be written as

$$\frac{dL}{dt} = \frac{B_{tot}}{W(a_1 + a_2)} - \frac{a_3}{(a_1 + a_2)}\frac{dS}{dt} \,, \tag{4}$$

where

$$a_1 = \frac{3}{2}H_m; \quad a_2 = -\frac{\nu H_m L}{(1 + \nu \bar{s})}\frac{\partial \bar{s}}{\partial L}; \quad a_3 = -\frac{H_m L}{S} \,. \tag{5}$$

The expressions for $\bar{s}$ and $\partial \bar{s}/\partial L$ will be given in section 2.3, where the bed geometry is discussed.

Eq. (4) is the prognostic equation for the model. Although there is no spatial resolution it should be stressed that the incorporation of eq. (3) implies that the height-mass balance feedback is fully taken into account. In fact, as has been demonstrated in Oerlemans (2011; Figure 5.8), the model fairly accurately reproduces the hysteresis implied by an overdeepening. When the balance profile with height is linear, only the mean ice thickness enters the expression for the surface mass budget (see next subsection). So the fact that the ice thickness is not calculated as a spatial variable has no effect on the calculated climate-driven evolution of the glacier.

For a given bed topography, the mean bed slope depends on $L$. So for a concave bed, a retreating glacier will become thinner because of its reduced length and a steeper bed. The MGM thus captures the feedback between geometry, glacier thickness and mass budget.

Eq. (4) is a nonlinear equation, because $L$ appears implicitly in $B_{tot}$ as well as in $H_m$ and $\bar{s}$. However, it is not a stiff equation and a stable numerical solution can easily be obtained by integration with the explicit first-order Euler scheme. Tests have shown that for all applications in this paper a 1a time step is practical and adequate. Computational demands are negligible: a 1000 a simulation typically takes one second on a laptop.

## 2.2 The mass budget

Surface mass balance measurements are not available for Monacobreen, and therefore the same strategy is followed as in the study of Abrahamsenbreen (Oerlemans et al., 2011). Based on existing mass balance observations on glaciers in northwest Spitsbergen filed at the World Glacier Monitoring Service (Zürich), the balance rate $\dot{b}$ is taken as a linear function of altitude, i.e.

$$\dot{b} = \beta(h - E) \,, \tag{6}$$

where $E$ is the equilibrium-line altitude (used as climatic forcing) and $\beta$ is the balance gradient. From the measurements it appears that a characteristic value is $\beta = 0.0045$ m w.e. m$^{-1}$ (Oerlemans and Van Pelt, 2015). The equilibrium-line altitude is used as the parameter to impose climate change to the glacier model. The total mass loss or gain of the flowband is found by integrating the balance rate over the glacier length:

$$B_s = \beta W \int_0^L [H(x) + b(x) - E]dx = \beta(H_m + \bar{b} - E)WL , \qquad (7)$$

where $\bar{b}$ is the mean bed elevation of the glacier (note that this quantity as well as $H_m$ depends on the glacier length).

The tributary glaciers / basins are treated in a simple way, in which the surface geometry is fixed. This is justified because the tributaries have larger slopes and therefore a weaker altitude - mass balance feedback. The tributaries are assumed to be in an equilibrium state, and therefore only supply mass to the main stream when they have a positive budget (which is the case when the mean surface elevation is less than $E$). Each tributary glacier is described as a basin with a tilted trapezoidal shape. The basin has length $L_y$ and width $w(y) = w_0 + qy$ , where $y$ is a local coordinate running from the lowest part of the basin ($y = 0$) to the highest point of the basin ($y = L_y$). The surface elevation is taken as $h(y) = h_0 + sy$, where $s$ is the surface slope. Each of the ten individual basins is thus characterized by five parameters: $L_y, w_0, h_0, s, q$. Note that the basins can become narrower when going up ($q < 0$) are wider ($q > 0$). The values of the geometric parameters were all estimated from the topographical map referred to in section 1, and are listed in Table 2. Since the interest is in the bulk mass budget of a basin, the uncertainties in the geometric parameters have a limited effect only. The total mass budget $B_i$ of basin $i$ is found to be

$$B_i = \beta \int_0^{L_y}(b_0 + sy - E)(w_0 + qy)dy = \qquad (8)$$

$$= \beta \left[ w_0(b_0 - E)L_y + \frac{1}{2}\{sw_0 + (b_0 - E)q\}L_y^2 + \frac{1}{3}sqL_y^3 \right] .$$

The final term in the mass budget is the calving rate. The calving rate is assumed to be proportional to the water depth $d$, so the calving flux can be written as

$$F = -c\, d\, WH_f, \qquad (9)$$

where $H_f$ is the ice thickness at the glacier front, and $c$ is the 'calving parameter'. The type of calving law formulated in eq. (9) has been suggested by, among others, Brown et al. (1982), Funk and Röthlisberger, 1989; Pelto and Warren (1991), Björnsson et al. (2000). There are many mechanisms leading to the production of icebergs (for a comprehensive review, see Benn et al., 2007). In recent years numerical models have been developed that simulate the calving process in great detail (e.g. Krug et al., 2014). Results from such models provide insight into the details of the calving process, but cannot be simply used to formulate a more general calving law for use in large-scale models of tidewater glaciers. In the present study the interest is in the global dynamics of a glacier system and its evolution on larger time scales. It is therefore reasonable to assume that mass loss by calving is generally larger when the glacier front is in deeper water.

The thickness at the glacier front is not explicitly calculated and therefore has to be expressed in the mean ice thickness and/or glacier length. As for Hansbreen, the following parameterization is used

$$H_f = max\{\kappa H_m; \delta d\} , \qquad (10)$$

where $\delta$ is the ratio of water density to ice density, and $\kappa$ is a constant giving the ratio of the frontal ice thickness to the mean ice thickness. So according to eq. (10) the ice thickness can never be less then the critical thickness for flotation.

The use of eq. (10) allows the model glacier to undergo a smooth transition between a land-based terminus and a calving front, which is a prerequisite for long-term simulations in which a model glacier should have the possibility to grow from zero volume to a long calving glacier, and backwards. Recent model studies with more comprehensive numerical models have focused on simulating and explaining the short-term (seasonal to decadal) fluctuations in calving fluxes and glacier front behaviour (e.g. Otero et al. 2017; Todd and Christoffersen, 2014). It should be stressed that the parameterization in the MGM is not meant to simulate such short-term fluctutations, but attempts to quantify the calving flux as a long-term

component of the total mass budget. With respect to the use of comprehensive numerical models it should be noted that
validation against long-term fluctuations of individual glaciers (including the LIA maximum stand) has not yet been
attempted / published.

At Hansbreen, Polish scientists have produced the longest series of calving flux measurements for any tidewater glacier in
Svalbard (Blaszczyk et al., 2009; Petlicki et al., 2015). There are certain similarities between the calving fronts of Hansbreen
and Monacobreen. The water depth around the current location of the calving front is in the same range, and both glaciers
exhibit a seasonal fluctuation of the calving front position of comparable magnitude (see also Mansell et al., 2012). There are
no systematic observations of the calving flux at Monacobreen that allow to determine a value of $c$. Therefore in this study
the same parameter values are used as found for Hansbreen, namely $c = 1.15\, a^{-1}$ and $\kappa = 0.4$.

## 2.3 Geometry of the main stream

The bed of Monacobreen has never been mapped. Airborne radio-echo sounding in the past has revealed strong internal
reflections, and it appeared to be impossible to map the bed of the glacier (Dowdeswell et al., 1984). Until today, the
thickness of Monacobreen is unknown. Methods have been developed to estimate ice thickness from the surface topography,
with or without information about the surface mass balance distributions (Farinotti et al., 2017). At the present state of the
art, these methods do not yet appear to be reliable for the larger glaciers in Spitsbergen. Input data is too uncertain and the
condition of near-equilibrium, needed to estimate the balance flux, is not met. The fact that time scales are large and surge
behaviour is present, complicates the application of ice-thickness models substantially. In addition there is a large
uncertainty about the sliding regime. A few test calculations done by the author revealed that estimated ice thicknesses are
much too large for the few glaciers on Spitsbergen for which bed elevation data exist.

However, the surface elevation of Monacobreen is quite smooth and regular (Fig. 3), and it is unlikely that there are
significant overdeepenings or major steps in the bed. For the nearby glaciers Kronebreen and Kongsvegen, a compilation of
bedrock data from various sources was published by Hagen and Saetrang (1991), with a further analysis in Melvold and
Hagen (1998). Kronebreen descends from the same plateau as Monacobreen, albeit in the other direction. The bed profile of
Kronebreen can be represented reasonably well with a function of the form $b_h \exp\left(-x/\lambda\right)$, where $b_h$ is the bed height at the
glacier head ($x = 0$) and $\lambda$ is the length scale that determines how fast the bed drops off along the glacier flowline (Van
Dongen, 2014).

Here a similarity between the bed profiles of Kronebreen and Monacobreen is assumed. This is based on the fact that these
glaciers share a number of characteristics (long calving glacier, fairly smooth topography, located in a similar geological
setting). The bed profile of the main stream of Monacobreen is therefore formulated as:

$$b(x) = b_a + b_h \exp\left(-x/\lambda\right) \ . \tag{11}$$

The value of $b_a$ (negative for a tidewater glacier) determines the bed height for $(x \to 0)$, and the bed elevation at the head of
the glacier now is $b_a + b_h$. The mean bed slope over the length of the glacier, needed in eq. (5), thus becomes

$$\bar{s} = \frac{b_h\left[1 - \exp\left(-\frac{L}{\lambda}\right)\right]}{L} \ . \tag{12}$$

The change of the mean bed slope with $L$, also needed in eq. (5), is

$$\frac{\overline{\partial s}}{\partial L} = -\frac{b_h\left[1 - \exp\left(-\frac{L}{\lambda}\right)\right]}{L^2} + \frac{b_h \exp\left(-\frac{L}{\lambda}\right)}{\lambda L} \ . \tag{13}$$

Finally, the mean bed elevation over the glacier length, needed in eq. (7) is found to be

$$\bar{b} = b_a + \frac{b_h \lambda \left[1 - \exp\left(-\frac{L}{\lambda}\right)\right]}{L} . \tag{14}$$

The bathymetry of Liefdefjorden is well known (e.g. Hansen, 2014). The water depth varies considerably, but is mostly between 50 m and 200 m (far away from the glacier in the fjord). It is therefore appropriate to use $b_a = -175$ m. The value of $\lambda$ is then determined by requiring that the water depth at the glacier front is close to the observed value of about 75 m (averaged over the width of the glacier). The resulting bed profile is shown in Fig. 3. Note that with this choice of parameters

calving occurs whenever the glacier is longer than about 28 km.

With the observed surface elevation and the parameterized bed profile, the corresponding mean ice thickness is 247 m. This is reproduced by the model when $\alpha$ in eq. (3) is set to 1.70 m$^{1/2}$. This is a fairly small value compared to the measured value for Hansbreen (3.0 m$^{1/2}$), but somewhat larger than that found for Kronebreen (1.43 m$^{1/2}$; Van Dongen, 2014). Apparently both Kronebreen and Monacobreen have beds that provide a relatively low resistance to ice flow. The parameter

values derived above are also listed in Table 1.

### 2.4 Imposing surges

Monacobreen surged from 1991 till 1997. The duration of surges of the larger glaciers in Svalbard is rather long, and Monacobreen is no exception. The velocity field and front positions during the surge have been documented in detail from analysis of a large number of satellite images (Mansell et al., 2012). During the surge the glacier front advanced by about 2

km. Compared to the glacier length, this is not much. It is likely that the LIA maximum stand as mapped in 1907 occurred at the end of a surge. This would suggest that the surge cycle of Monacobreen is about 100 years.

In the model the surge function is prescribed as (Oerlemans, 2011):

$$S(t) = 1 - S_0(t - t_0)e^{-(t-t_0)/t_s} \quad \text{for } t \geq t_0 , \tag{15}$$

The surge starts at $t = t_0$, and $t_s$ is the characteristic time scale of the surge. Denoting the length of the surge cycle by $T_s$, the

values of $t_0$ are regularly spaced with intervals $T_s$. $S_0$ determines by how much the mean ice thickness is reduced at the end of the surge.

Surges as reproduced in the model are shown in Fig. 4. It appears that the observed amplitude and duration of the surge can be matched well with $S_0 = 0.027 \ a^{-1}$ and $t_s = 8$ a. A surge cycle of 100 years has been prescribed ($T_s = 100$ a). The equilibrium-line altitude was constant and set to 619 m, because this produces a glacier length of about 40 km. When a surge

begins, the mean ice thickness (and thus surface elevation) immediately starts to decrease. This is a direct consequence of mass conservation. The lower surface leads to a negative net balance that peaks at the end of the surge. The curves for the net balance and mean ice thickness are not identical, as the glacier area also changes (the net balance is the total mass budget divided by the glacier area). The change in the net balance also involves a contribution from the changing calving rate, but for the present geometric set up (the water depth increase only very slightly at the glacier front) this is an insignificant effect.

The maximum perturbation of the net balance is only -0.13 m w.e. a$^{-1}$. This is much less than for Abrahamsenbreen, where it was found to be -0.5 m w.e. a$^{-1}$ (Oerlemans and Van Pelt, 2015). The difference stems from the fact that the relative surge amplitude (maximum frontal advance divided by glacier length) for Abrahamsenbreen is more than twice that for Monacobreen. In view of the small effect on the net balance, it is likely that the effect of regular surging on the long-term response of Monacobreen to climatic change is limited. This will be illustrated further in section 5.2, where an integration

with surging is compared to an integration without surging.

## 3 Basic sensitivity and response time

According to the map in Hagen et al. (1993), the estimated equilibrium-line altitude (ELA) in the region of Monacobreen is 500 to 600 m a.s.l., with a tendency to lower values when going in northwesterly direction. The estimate is based on an assumption of balance, i.e. a situation in which the glaciers would be in equilibrium. This is currently not the case and the equilibrium line over the past decades has certainly been higher. In the model the value of $E$ is taken the same for the entire domain, except for basins 1, 2 and 3, where $E$ is perturbed by -100, -100, and -50 m, respectively. This is done to take the decline of the ELA as mapped in Hagen et al. (1993) into account. In fact, without the lowering of the ELA the net mass budget of basin 1 (Seligerbreen) would be negative and the tributary could never supply mass to Monacobreen (as it did until recently). Altogether, the ELA map of Hagen et al. (1993) appears to be consistent with the mass budgets of the tributaries.

The evolution of Monacobreen for stepwise forcing of the equilibrium line is shown in Fig. 5. As can be expected in view of the very small slope, it turns out that the climate sensitivity of the glacier is very large. Changing $E$ from 775 to 575 m makes the glacier grow by 24 km. Bringing back the equilibrium line to 600 m (in two steps) does not reveal a sign of hysteresis (as would probably occur when there would be an overdeepening in the bed). The adjustment to a changing climate apparently takes a long time: the e-folding response time is about 250 a.

The corresponding components of the mass budget are shown in Fig. 5. To facilitate a comparison, all components are expressed as a specific balance. The important role played by the tributary glaciers / basins becomes clear immediately. Only at $t = 500$ CE, when $E$ drops instantaneously by 200 m, does the main stream have a positive balance. This does not last very long, however, because the increasing glacier length soon leads to a much larger ablation zone. When the glacier length is of the order of 40 km, as it is today, the negative net surface balance of the main stream and the calving flux contribute roughly equally to the compensation of the net input from the tributaries.

The time scale of about 250 years can be compared with an estimate of the much used volume time scale $\tau_J$ defined by Jóhannesson et al. (1998):

$$\tau_J = -H^*/\dot{b}_{x=L} \, , \tag{16}$$

where $H^*$ is the maximum ice thickness and $\dot{b}_{x=L}$ is the balance rate at the glacier front. Using 350 m as a maximum ice thickness and $-2.5$ m a$^{-1}$ as a typical balance rate yields a value of about 140 a. However, as demonstrated by Oerlemans (2001) and confirmed by Leysinger Vieli and Gudmundsson (2004) with a higher order numerical glacier model, for glaciers with small slopes the altitude-mass balance feedback makes the time scale considerably longer. Since Monacobreen has a very small average slope (0.027), the value of $\sim 250$ a is a plausible one.

## 4 Climatic forcing

A simulation of the Holocene evolution of Monacobreen requires an appropriate climatic forcing function. Fortunately, a ELA history for a small valley glacier on Mitrahalvøya, which is only 25 km west of the central part of Monacobreen, has been reconstructed from lake sediments (Røthe et al., 2014). This glacier, Karlbreen, currently has an area of about 2 km$^2$ and the glacier head is at about 500 m a.s.l. This is an important height because it puts a lower limit on the ELA in times that there was no glacial activity (in this case the ELA has to be above the mountain top). The glacier drains through a series of three lakes and an extensive sediment analysis has provided a full history of glacial activity. By using former glacier stands, Røthe et al. (2014) were able to convert sediment parameters to ELA values, assuming an equilibrium between glacier size and ELA. According to Røthe et al. (2014), there was very little glacial activity during the period 9200 to 3500 BCE, implying that the equilibrium line was mostly above 500 m a.s.l. After the Holocene Climatic Optimum there has been a

long-term trend towards lower values of the ELA, with significant fluctuations superposed on this. Since a glacier like Karlbreen probably has a response time scale of the order of 20 to 30 a, the reconstruction is bound to be less accurate for shorter time scales. Therefore the reconstruction for Karlbreen is combined with ELAs esitmated from meteorological observations at Longyearbyen / Svalbard airport since 1899 (Førland et al., 2011).

There is a distinct west-east gradient in the ELA in northwest Spitsbergen (Hagen et al., 1993). The ELA rises in eastward direction, mainly due to lower precipitation rates. Therefore, the *absolute* reconstructed ELA values cannot be used to force the model for Monacobreen, so the ELA perturbation $E_R$ relative to a reference value $E_0$ is used. The ELA is now formulated as

$$for\ t < 1899:\ E(t) = E_0 + E_R(t)\ . \tag{17a}$$

$$for\ t \geq 1899:\ E(t) = E_1 + E_M(t)\ . \tag{17b}$$

Here $E_M(t)$ is the ELA perturbation obtained from the meteorological data. The constants $E_0$ and $E_1$ need not necessarily be the same, because it is unknown how precisely the meteorological observations connect to the past. These constants will therefore be used as tuning parameters.

The relation between the ELA and temperature / precipitation is based on calculations with an energy balance model, as described in Van Pelt et al. (2012) and used in Oerlemans and Van Pelt (2015). The sensitivities are $\partial E / \partial T = 35$ m K$^{-1}$ and $\partial E / \partial P = -2.25$ m %$^{-1}$, where $T$ and $P$ are perturbations of the annual mean temperature and precipitation. It should be noted that the value of $\partial E / \partial T$ is rather small compared to values found for glaciers in a midlatitude alpine setting, which are of the order of 100 m K$^{-1}$. This stems from the fact that summer temperature anomalies over Spitsbergen (and in general over the Arctic region) are much smaller than mean annual temperature anomalies. Since summer temperature determines to a large extent the ELA perturbation, the net effect is that the sensitivity to an annual temperature anomaly is relatively small (for a further discussion on this point, see Van Pelt et al. , 2012).

As shown in Fig. 6a, the variation of reconstructed ELA values from mid-Holocene times until today have a typical range of 200 m. If this would solely be a temperature effect the drop in ELA since the mid-Holocene would correspond to a 5.7 K decrease in temperature (according to the sensitivity referred to above). This is more than reconstructions of mid-Holocene warmth in the Arctic actually suggest, which are in the 2 to 4 K range (e.g. CAPE, 2006; Bradley, 2016; Axford et al., 2017). However, there is also a direct effect of changes in orbitally-driven insolation variations. The differences in summer insolation between mid-Holocene and present day are between 5 and 10%, depending on the precise location and definition of the summer season (Berger and Loutre, 1991). The increased insolation certainly caused higher melt rates in the mid-Holocene, and thereby a higher equilibrium line.

## 5 Holocene evolution of Monacobreen

The Minimal Glacier Model constructed in the previous sections has been forced with the ELA reconstruction discussed above. Information on former stands of Monacobreen is limited, but the existing data points can nevertheless be used to calibrate and test the model. For a succesful simulation, in which the simulated glacier length matches the observed record, two conditions have to be fulfilled, namely, (i) the MGM should have sufficient dynamics to respond in a realistic way to climate forcing, and (ii) the forcing function (the ELA reconstruction) is accurate enough to make the glacier grow and shrink at the right times. It is not at all trivial, even with optimal values of $E_0$ and $E_1$, that the maximum glacier stand comes at the right time.

In section 5.1 the best possible simulation will be discussed (the reference simulation). In section 5.2 sensitivity tests are described in which parameters are varied or processes are switched off.

## 5.1 Reference simulation

The ELA reconstruction from the Karlbreen basin is available from the year 2155 BCE, but the model integration starts a bit earlier to have an equilibrated state at this time. For the final outcome the precise choice is not important, because the response time of the glacier is much shorter than the period of integration. Surging is included, and the duration of the surge cycle is set to 100 a. The strategy is simple: trying to adjust the values of $E_0$ and $E_1$ until the LIA maximum stand ($L = 43.3$ km) and the 1997 maximum front position at the end of the surge ($L = 40.7$ km) are reproduced. This turned out to be the case for $E_0 = 584$ m a.s.l. and $E_1 = 627$ m a.s.l., which are plausible values. There appeared to be no need to adjust the calving parameter.

The corresponding evolution of the glacier length is shown in Fig. 6. Due to declining summer insolation, at the end of the Holocene Climatic Optimum the arctic climate cools and the equilibrium line drops. For Monacobreen the ELA is then around 830 m and the length of the glacier is about 15 km. From 1700 BCE the ELA decreases markedly, and the glacier grows to a size of about 30 km and becomes a tidewater glacier. During a few centuries BCE, the ELA fluctuates between 650 and 700 m, until it drops by another 100 m in the first century CE. In this range of ELA values Monacobreen is quite sensitive, and the 100 m drop in the ELA is sufficient to make the glacier grow to its current size. The lowest ELA values for the entire period are during the 19th century and this then leads to a maximum glacier stand around 1900 (in combination with the surge). The rise of the ELA during the 20th century leads to a 3.5 km retreat of the glacier front until the last surge starts in 1991. This surge coincides with a sharp increase in the ELA of another 100 m, and as a consequence the retreat of the glacier front after the surge is almost twice as fast as after the previous surge. In Fig. 6b a close up is given of the past few hundred years, including the observed glacier front positions (in terms of glacier length).

At this point it should be recalled that the tuning procedure is straightforward: four calibration parameters $\{E_0, E_1, S_0, t_s\}$ have been used to match: (i) the LIA maximum stand, (ii) the glacier stand at the onset of the surge, (iii) the amplitude of the surge, and (iv) the time scale of the surge.

The simulated position of the glacier front in 2016 is rather close to the observed position without any further tuning. Apparently this result can be obtained with a calving parameter that is constant in time. This is in agreement with the analysis of Mansell et al. (2012), who did not find significant fluctuations in the calving rate during the surge. Altogether, the simulated evolution of Monacobreen during the Holocene is in good agreement with the observational evidence. A great deal of this result is probably on the account of the realistic ELA reconstruction from the Karlbreen basin (Røthe at al., 2014).

The components of the mass budget corresponding to the model integration of Fig. 6 are shown in Fig. 7. For reference the glacier length is also shown. A distinction is made between the total contribution from the tributaries, the surface balance of the main stream and the calving flux. All components are shown as a specific balance, i.e. expressed as a specific net loss or gain averaged over the area of the main stream. The contribution from the tributaries is always positive, and the surface balance of the main stream is always negative. Since these two components are entirely determined by changes in the ELA, which are always the same over the enire domain, it is not surprising that the fluctuations are highly correlated (note that the area of the main stream varies significantly). Around 500 BCE Monacobreen becomes a tidewater glacier and the calving flux gradually becomes more important. During the past 1000 years the mass los by calving is of the same order as the surface

balance of the main stream. However, after the 1991-1997 surge the surface balance of the main stream has become much

more negative, and represents a greater loss than the calving. This is clearly due to the strong increase in the ELA.

## 5.2 Sensitivity tests

The reference balance gradient used in this study has been taken as $\beta = 0.0045$ m w.e. a$^{-1}$ m$^{-1}$. This is based on long-term obervations on a number of glaciers in western Spitsbergen. As discussed in Oerlemans (2001), balance gradients are smaller in dryer climates (e.g. typically 0.003 m w.e. a$^{-1}$ m$^{-1}$ on cold glaciers in the Canadian Arctic) and larger in wetter climates

(up to 0.01 m w.e. m$^{-1}$ on the the maritime glaciers of New Zealand and Patagonia). On many mid-latitude glaciers values of 0.007 m w.e. a$^{-1}$ m$^{-1}$ are found. Calculations with simple as well as more comprehensive models have show that the effect of varying $\beta$ on the equilibrium size of glaciers is not very large, whereas response times may change significantly (e.g. Oerlemans, 2001; Leysinger Vieli and Gudmundsson, 2004). Glaciers with larger balance gradients respond faster to climate change, as can also be seen in eq. (16). Against this background simulations with different values of $\beta$ were carried out. The

earlier findings were confirmed: also for Monacobreen a larger value of $\beta$ implies a somewhat larger glacier for a given ELA as well as a shorter response time for a change in the ELA.

Since Monacobreen was in a slightly warmer climate than today during most of the Holocene, a case with a somewhat larger values of $\beta$ is an appropriate one to discuss. In Fig. 8 the reference run described in section 5.1 is compared with a simulation with $\beta = 0.0065$ m w.e. a$^{-1}$ m$^{-1}$. Clearly, the larger balance gradient leads to a slightly larger glacier. Averaged

over the last 1000 years the difference in length is 2.8 km. In order to match the maximum glacier stand as well as the glacier length at the start of the surge, the model has to be re-calibrated. This is accomplished by adjusting the values of $E_0$ and $E_1$ by +16 m and +2 m, respectively. The corresponding simulated glacier length is also shown in Fig. 8. It is obvious that after recalibration the differences with the original simulation are small.

The dashed curve in Fig. 8 shows the result of a calculation in which surging was switched off. Because the surge amplitude

is quite small (see discussion in section 2.4), it is not surprising that the effect of surging on the long-term evolution of Monacobreen is limited. In the absence of surging the mean glacier length (e.g. averaged as a 100-a running mean value) is slightly larger, the difference being about 0.5 km over the past 1000 years. Varying the surge amplitude reveals that the effect is roughly proportional to this amplitude (not shown). For a larger balance gradient the effect of surging on the long-term mean glacier length is somewhat larger, but still small compared to the total length.

In another set of experiments the calving parameter $c$ was varied (Fig. 9). In the first run the calving was set to zero. This has a profound effect on the late-Holocene evolution of the glacier. The model predicts that without mass loss by calving the glacier would at present be 15 km longer. Halving the calving parameter also makes the glacier larger, but now a recalibration has been carried out by simply adjusting the ELA. Raising the equilibrium line by just 15 m appears to be enough to match the observed length record again (red curve in Fig. 9). Doubling the calving parameter, on the other hand,

requires a 35 m drop of the equilibrium line to match the observations again. When looking further back in time the recalibrated runs for different calving parameters show somewhat different results, but the overall pattern of glacier evolution during the late Holocene appears to be a robust finding. It is clear that the forcing function dominates the evolution, and that on the longer time scales considered here the glacier mechanics are 'slaved' by the external forcing.

## 6 Future evolution of Monacobreen

Recent trends and future climate change in Svalbard has been examined in detail by Fjørland et al. (2011). There is broad agreement between temperature trends obtained by downscaling results from global climate models (forced with greenhouse gas emissions) and observations at Svalbard Airport for the period 1912–2010. Projections for the Svalbard region indicate a future warming rate up to year 2100 three times larger than the observed rate during the past 100 years. For precipitation, the long-term observational series show a modest increase over the past 100 years and projections indicate a further increase up

to year 2100. In the present study the projected changes in temperature and precipitation for northwest Spitsbergen are used to estimate the future trend in the ELA.

For the B2 emission scenario (IPCC, 2013), the rate at which annual temperature and precipitation increases in the ensemble-mean projection is fairly linear until about 2090 and levels off somewhat afterwards (Fjørland et al., 2011). Combining the mid-B2 annual temperature and precipitation trends with the ELA sensitivities discussed in section 3 yields

$dE/dt = 1.86$ m a$^{-1}$. Against this reference the possible future evolution of Monacobreen was studied by imposing values of $dE/dt$ of 0, 2, 4 and 6 m a$^{-1}$. It should be noted that these values have also been used for glaciers in other parts of the world (e.g. in the Alps; Oerlemans et al., 2017). Although warming is larger in the polar regions (the 'polar amplification'), this is not so much reflected in the rise of the equilibrium line. The reason is that summer temperatures increase much less than annual mean temperatures. In the calculation of the ELA sensitivities this has been taken into account. However, since

the climate change scenarios are formulated as a change in the ELA, the possibility remains to convert meteorological variables into changes in the ELA for different sensitives.

When making projections of future climate change scenarios the outcome depends on the choice of the reference period. Starting from a warm year (e.g. 2015, ELA = 809 m) and increasing the ELA by a certain amount will give a very different result from starting in a cold year (e.g. 2014, ELA 668 m). Therefore the reference ELA should be a mean value over a

longer period. Moreover, it is unclear to what extent the very high ELA values since 2000 represent an expression of natural variability on the decadal time scale, or are a direct response to greenhouse-gas induced warming. To deal with this uncertainty, two 30-a reference periods were used to define the ELA perturbation associated with the projected climate change: (i) 1987-2016, i.e. a recent 30-yr period; and (ii) 1961-1990 as the last official period to define the climatology. The resulting eight projections of glacier length are shown in Fig. 10a. The integrations are extended until 2200, and the ELA-

perturbation is kept fixed after 2100. The curves immediately make clear that typically half of the response to 21st century warming will come after 2100.

For reference period (ii) and no climate change ($dE/dt = 0$), Monacobreen would hardly retreat. Note that this scenario actually implies that ELA values return to significantly lower values than observed over the last two decades. In contrast, for reference period (i) and no climate change, in the year 2100 the glacier front would have retreated by 3.5 km.

For the strong-warming scenario with $dE/dt = 4$ m a$^{-1}$ and reference period (i), the model predicts a glacier retreat of about 9 km by the year 2100. At this point the glacier would be grossly out of balance, since the ELA is then at 1038 m a.s.l., which is above most of the accumulation basin. In 2100 the volume of Monacobreen would be about 60% of the present-day volume (Fig. 10b). For the extreme warming scenario $dE/dt = 6$ m a$^{-1}$ and reference period (i), Monacobreen would virtually have disappeared within two centuries.

If the Paris-agreement would become reality, the mid-B2 scenario with $dE/dt = 2$ m a$^{-1}$ is perhaps the most likely one. In this case the front of Monacobreen would retreat by 3.5 to 5.5 km in the coming 80 years, depending on the choice of reference period. The glacier volume would have been reduced by 20 to 30 % with respect to the current volume.

**7 Discussion**

In this paper it has been demonstrated that the conceptual simplicity of the MGM makes it possible to study the response of a large tidewater glacier to climate change in a transparent way. Calibration turned out to be a straighforward procedure, in which three characteristic glacier front positions could be reproduced accurately (LIA maximum stand, front position at the end of the recent surge, present-day front position). Here the quality of the reconstructed ELA history from Røthe at al. (2014) certainly plays an important role. The approach of defining a main stream that is fed by tributary glaciers / basins seems to work well. The tributary basins were included in a passive way. In fact, the basins were assumed to have a fixed geometry and to be in balance with the prevailing ELA. This should work well as long as the characteristic time scales of the basins are shorter than that of the main stream (perhaps for Louetbreen this is not quite the case). In principle all individual basins can be modelled with a MGM as well, implying the introduction of a response time and the local height - mass balance feedback. However, for the present study this was not considered appropriate given the little information available for these basins.

A significant result from this study is that surging has a limited effect on the long-term evolution of Monacobreen. The relative surge amplitude $A_s$, loosely defined as the maximum advance divided by the glacier length, is quite small ($\approx 0.05$). $A_s$ determines to a large extent by how much the mean surface elevation drops. The combined effect of a lower suface elevation and a larger area determines the mass-budget perturbation implied by the surge. It was found that for Monacobreen this perturbation amounts to $-0.13$ m w.e. $a^{-1}$ (when expressed as a change in the net balance rate of the main stream, see Fig. 4). It is interesting to compare these numbers with those of Abrahamsenbreen (Oerlemans and Van Pelt, 2015). For Abrahamsenbreen the surge amplitude is 0.14, and the perturbation of the mass budget about $-0.5$ m w.e. $a^{-1}$ .

It should be noted that the perturbation of the mass budget is solely determined by the redistribution of mass, not by the details of how this distribution actually takes place. When a glacier increases its length during a surge, the change in mean surface elevation is entirely dictated by the conservation of mass, not by the details of the surging mechanism. This implies that conclusions about the effect of surging on the long-term mass budget can be drawn even when the surging process itself is not fully understood.

Calving has a significant effect on the total mass budget of Monacobreen, but different values of the calving parameter do not change the qualitative evolution of the glacier during the Holocene very much. The range over which Monacobreen fluctuates is somewhat smaller for a larger calving parameter (the green curve in Fig. 9). This is understandable for a bed profile that slopes downward along the flowline, because the front of a growing glacier comes into deeper water and the mass loss by calving increases.

It has been observed that on shorter time scales details of the bathymetry may have significant effects on the calving rate and thereby on the position of a tidewater glacier front (e.g. Vieli et al., 2002). According to the measured bathymetry in the Liefdefjorden, these variations, with an amplitude of 10 to 50 m, are irregularly spaced and consist mainly of deposited moraines. It is unlikely that a similar bed would currently be present under Monacobreen with its very smooth surface, or existed in the fjord before the glacier started to advance in late Holocene times. Therefore it does not seem meaningful to include a map of the present-day bathymetry of the Liefdefjorden in one way or another. Probably, the smaller features of the bed profile do not matter too much for the glacier evolution on longer time scales, unless there are very marked jumps in bed or side geometry that could serve a pinning points. However, this does not seem to be the case.

The MGM has very simple dynamics: there is no spatial resolution and the mechanics are contained in a relation between length, mean ice thickness and mean bed slope, with simple formulations of the calving and surging processes. These formulations do not shed further light on the *nature* of the calving and surging processes, but the effects on the mass budget, and thus on the long term evolution of the glacier, are dealt with. At this point one may ask whether it would be posssible to

study the Monacobreen glacier system with a 3-dimensional higher order model and make a comparison with the simple approach taken here. Modelling the main stream would probably be feasible, but to deal with all the tributaries would require a large amount of data on the bed geometry which are not available and thus would have to be generated indirectly. Generating the surge cycle in a higher-order model, involving a coupling of ice mechanics and glacier hydrology, is another major difficulty to deal with. Perhaps at the present state of the art, it would be more realistic to strive for a hybrid model, in which a comprehensive model for the main stream is combined with MGM's for the tributary glaciers / basins. However, on longer time scales glacier mechanics are always slaved by the mass budget. The results from the recalibrated runs with different parameter values for balance gradient, surge amplitude and calving parameter support this view. It thus seems likely that a model with a comprehensive treatment of ice mechanics, when driven with the same climatic forcing and calibrated with the observed glacier stands, will produce a similar evolution of Monacobreen during the late Holocene.

## Acknowledgements

I am grateful to Torgeir Opeland Røthe (University of Bergen) for making the ELA-reconstruction data available, and to Øyvind Nordli (MET Norway) for supplying meterological data for Longyearbyen / Svalbard airport. The comments from two anonymous referees were very helpful in improving the paper – thank you!

This study has been sponsored by the Netherlands Earth System Science Centre (Utrecht, The Netherlands).

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

| Param. | value | meaning | reference |
|---|---|---|---|
| $\nu$ | 10 | eq. (3) - relation between ice thickness and slope | Oerlemans (2001) |
| $\alpha$ | 1.70 m$^{1/2}$ | eq. (3) - calibrated to give observed surface height | Map, Norsk Polarinstitutt |
| $\beta$ | 0.0045 m w.e. a$^{-1}$ m$^{-1}$ | balance gradient, observed on nearby glaciers | Oerlemans and Van Pelt (2015) |
| $b_a$ | -175 m | 'asymptotic' depth of fjord | Based on map Hansen (2014) |
| $b_h$ | 1100 m | note: $b_a + b_h$ is highest point of bed | Map, Norsk Polarinstitutt |
| $\lambda$ | 15000 m | calibrated to give observed water depth at front | Based on map Hansen (2014) |
| $\kappa$ | 0.4 | ice thickness at front (fraction of $H_m$) | Oerlemans et al. (2011) |
| $c$ | 1.15 a$^{-1}$ | calving parameter, as observed for Hansbreen | Oerlemans et al. (2011) |
| $S_0$ | 0.027 a$^{-1}$ | calibrated with amplitude of 1991-1997 surge | Mansell et al. (2012) |
| $t_s$ | 8 a | calibrated with observed duration of 1991-1997 surge | Mansell et al. (2012) |
| $\partial E/dT$ | 35 m K$^{-1}$ | based on energy-balance modelling | Van Pelt et al. (2012) |
| $\partial E/dP$ | -2.25 m %$^{-1}$ | based on energy-balance modelling | Van Pelt et al. (2012) |
| $E_0$ | 584 m | reference ELA for time < 1899, tuning parameter | tuning to length  record |
| $E_1$ | 627 m | reference ELA for time >1899, tuning parameter | tuning to length record |

**Table 1.** Overview of model parameters with corresponding references for more information.


| basin | $L_y$ (m) | $w_0$ (m) | $b_0$ (m) | $s$ | $q$ (m) |
|---|---|---|---|---|---|
| **T1** | 7000 | 2200 | 50 | 0.14 | -1.00 |
| **T2** | 3000 | 4800 | 200 | 0.23 | 0.0 |
| **T3** | 8500 | 3000 | 300 | 0.065 | 0.18 |
| **T4** | 3200 | 1500 | 550 | 0.11 | 0.71 |
| **T5** | 4700 | 3800 | 800 | 0.085 | 0.54 |
| **T6** | 4300 | 7500 | 900 | 0.11 | -0.65 |
| **T7** | 4700 | 1600 | 650 | 0.095 | -0.12 |
| **T8** | 7600 | 1800 | 500 | 0.078 | 0.82 |
| **T9** | 5700 | 4300 | 450 | 0.080 | 0.70 |
| **T10** | 3800 | 800 | 220 | 0.2 | 0.54 |

**Table 2.** Geometrical characteristics of the tributary basins of Monacobreen. Basin numbers as indicated in Fig. 2.



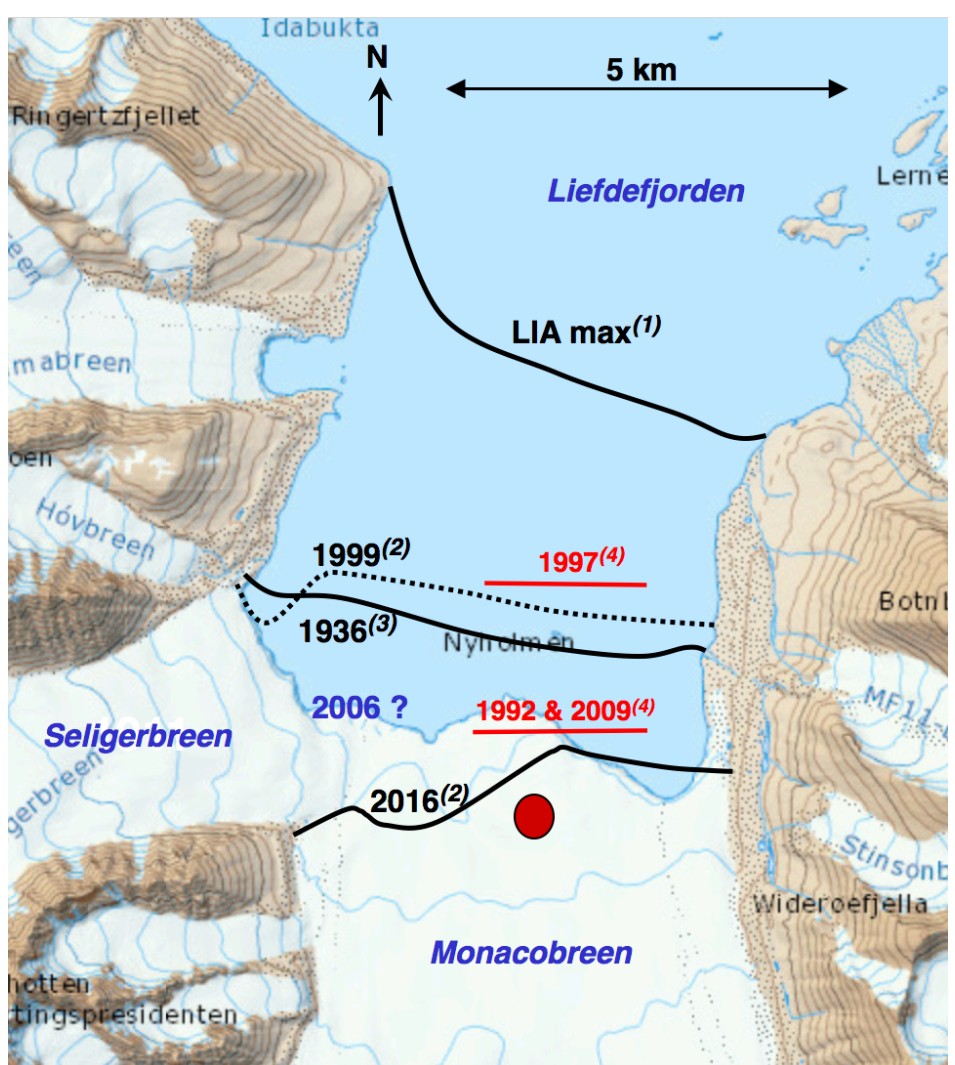

**Figure 1.** Front positions of Monacobreen in different years, drawn on a topographic map of the Norsk Polartinstitutt (http://toposvalbard.npolar.no) . The Little Ice Age maximum stand occurred around 1900 (Martin-Moreno et al., 2017), the 1999 and 2016 positions are from Landsat images (Pelto, 2017). The maximum front position related to the surge was in 1997 (Mansell et al., 2012). The 1992 position is just prior to the surge. In 2009 the glacier front had retreated back to this location. The red dot serves as a reference point for glacier length (L = 37.8 km).

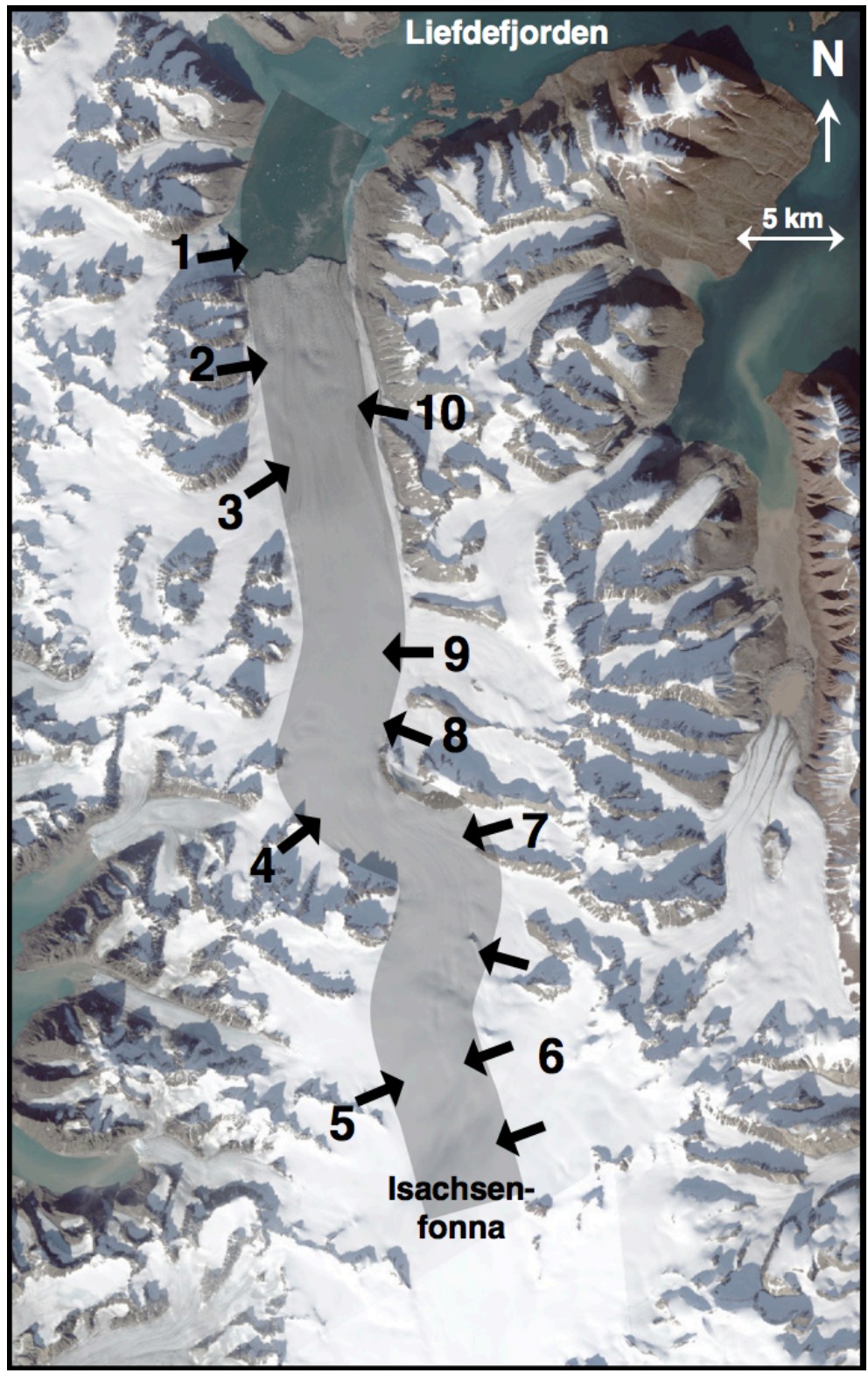

**Figure 2.** The 43 km long and 5 km wide flowband (grey) used to model Monacobreen, shown as an overlay on a Landsat image (©Norsk Polarinstitutt). The tributary glaciers / basins are numbered 1 to 10, and the corresponding geometric properties are listed in Table 2.

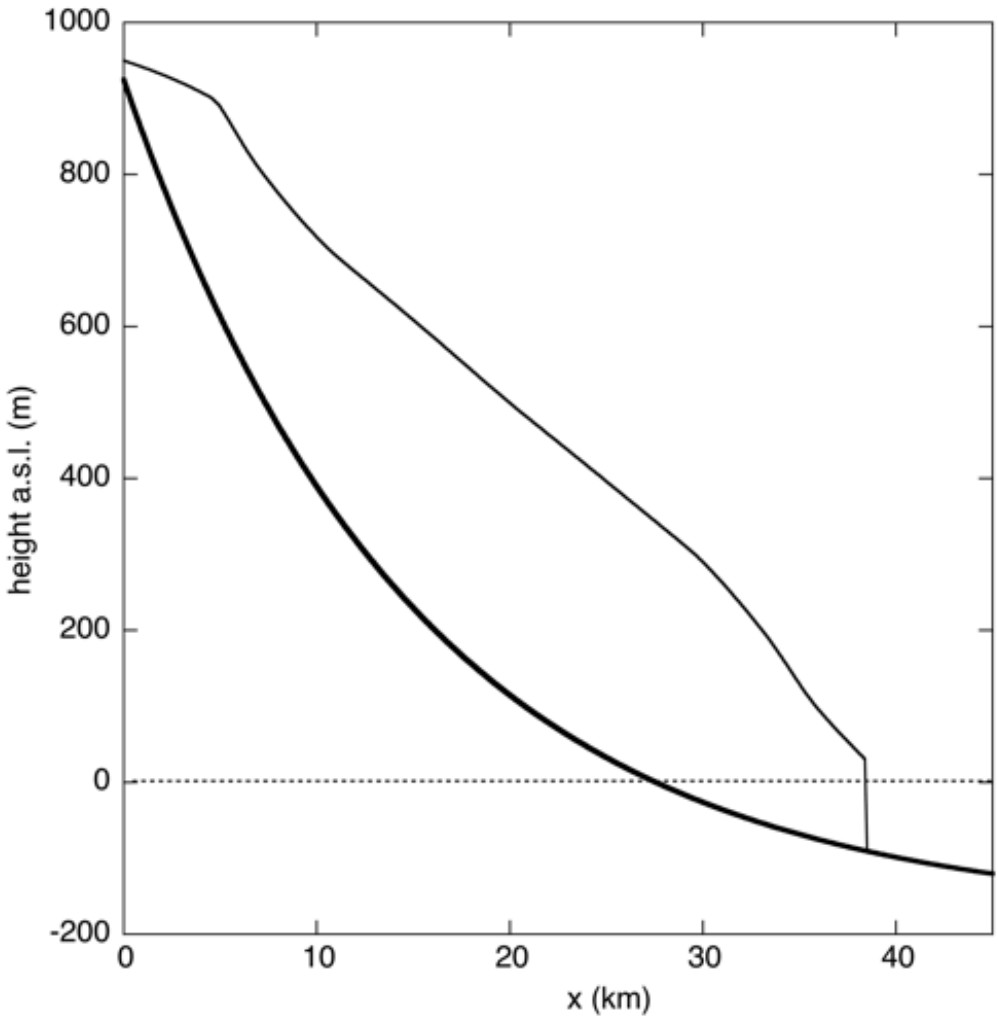

**Figure 3.** Upper curve: glacier surface elevation of Monacobreen along the flowband according to the topographic map ((http://toposvalbard.npolar.no). Lower curve: the bed profile as used in the model.

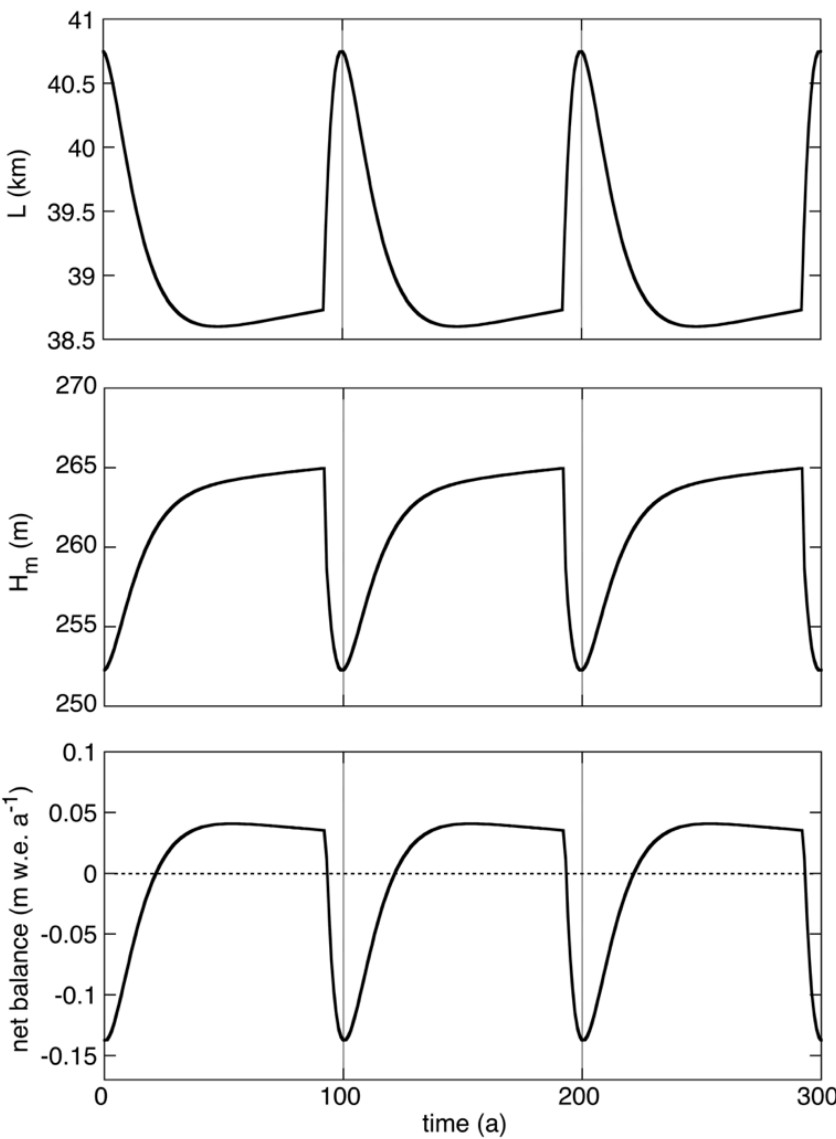


**Figure 4.** Illustration of the modelled surge cycle, as characterized by glacier length (upper panel), mean ice thickness (middle) and net balance (lower). The duration of the surge cycle has been set to 100 years.


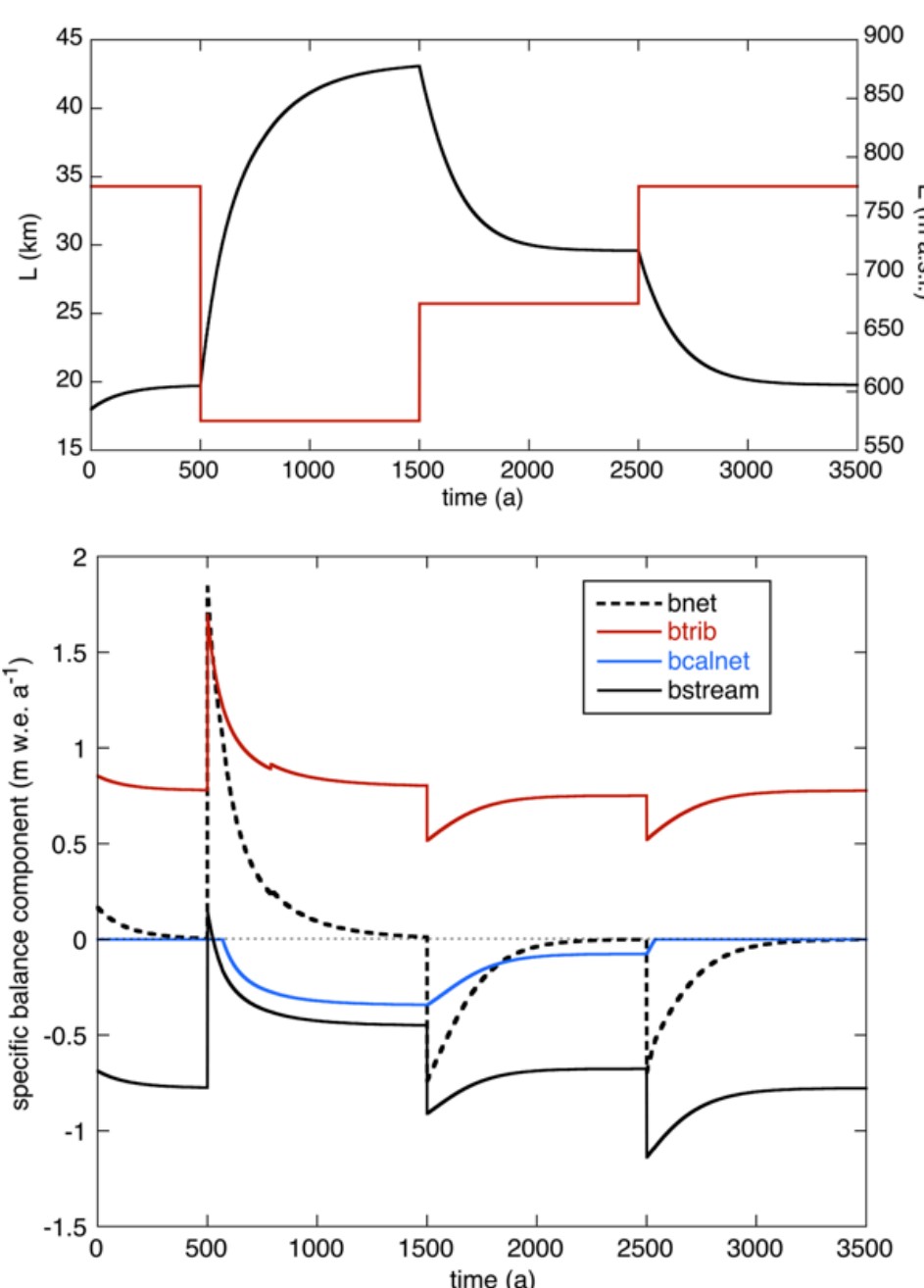

**Figure 5.** Upper panel: Evolution of glaciers length $L$ (black) for stepwise changes in the equilibrium-line altitude $E$ (in red). Lower panel: The corresponding components of the mass budget.


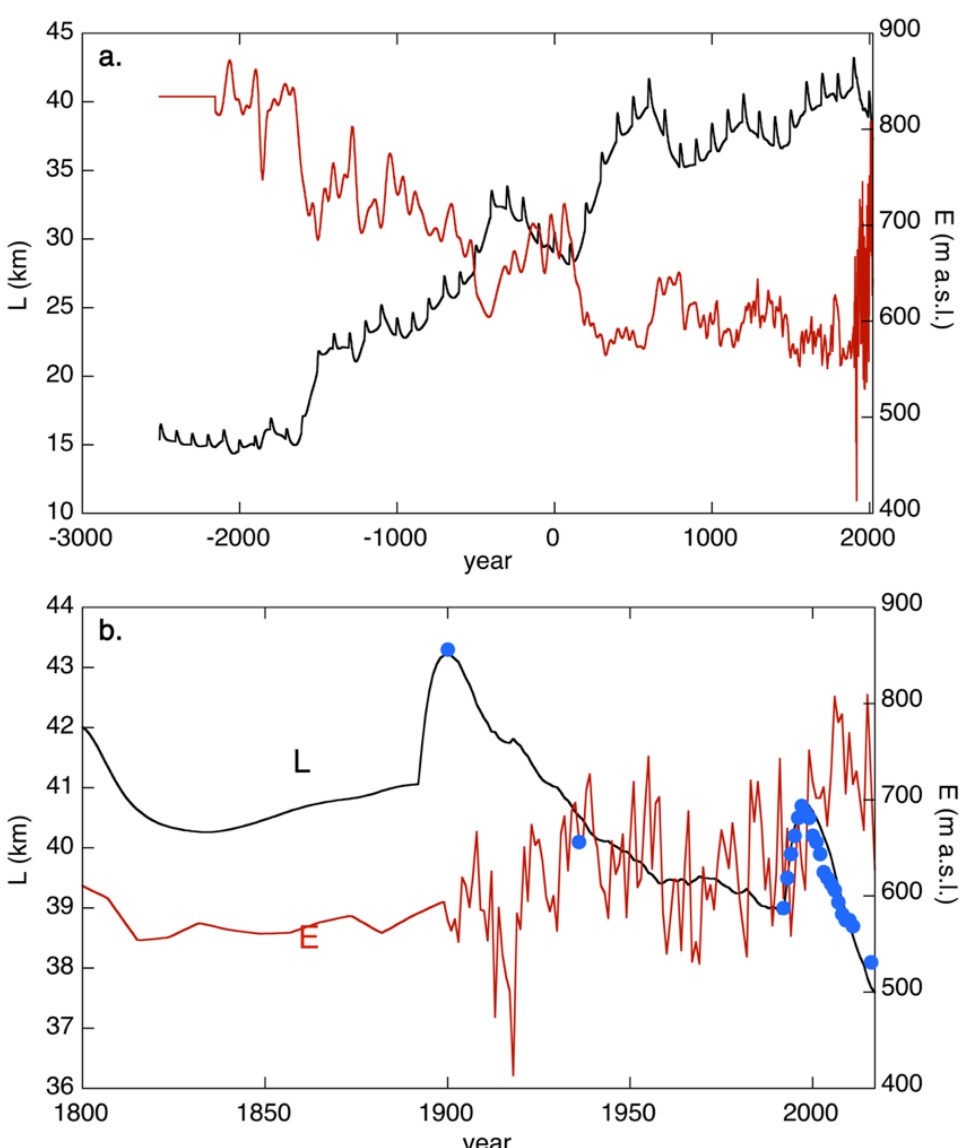

**Figure 6. a**: Evolution of glaciers length *L* (black) as a response to the reconstructed equilibrium-line altitude *E* (in red); **b**: close-up of the period 1800 - 2016. Observed glacier stands are shows as blue dots.

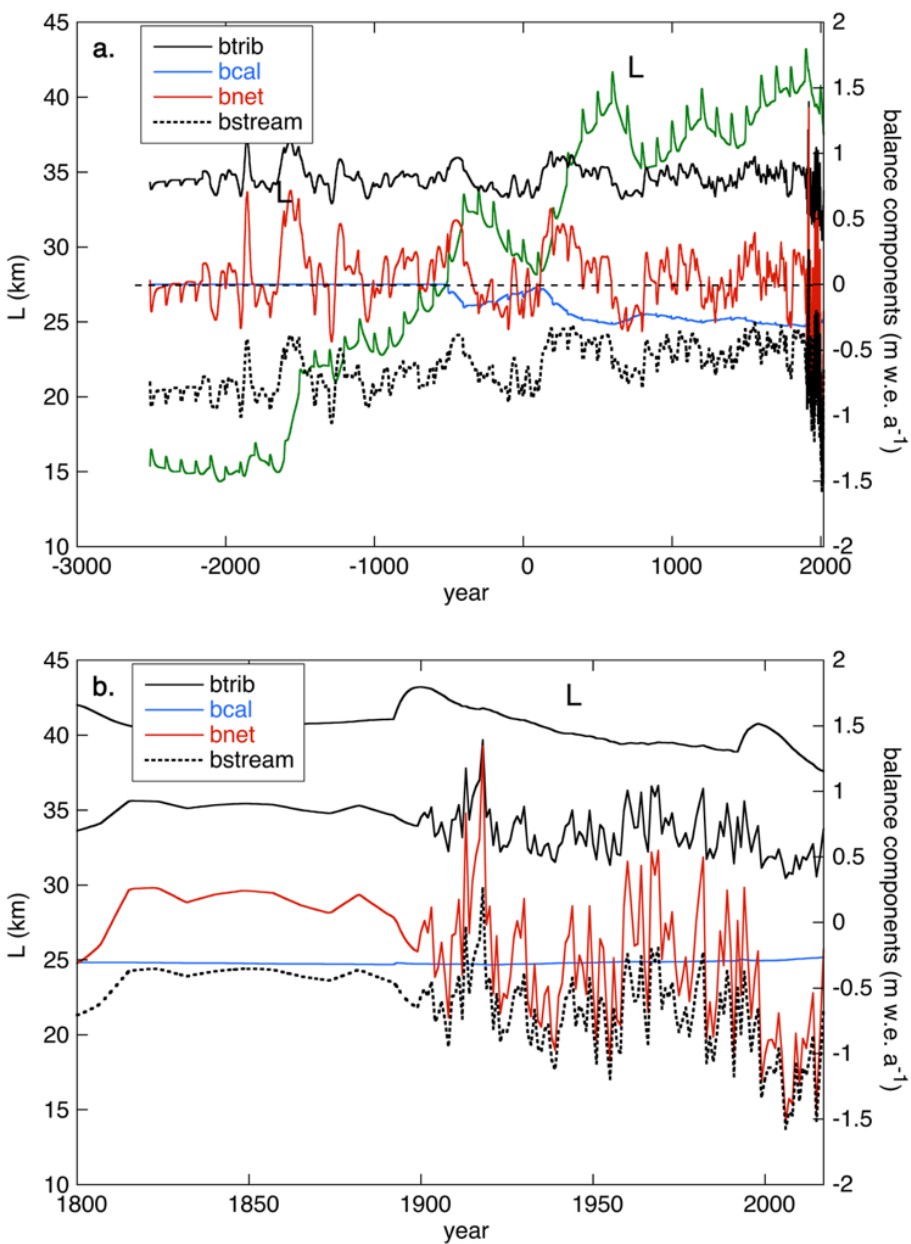

**Figure 7.** The components of the mass budget corresponding to the simulation shown in Fig. 6, expressed as a specific balance rate.


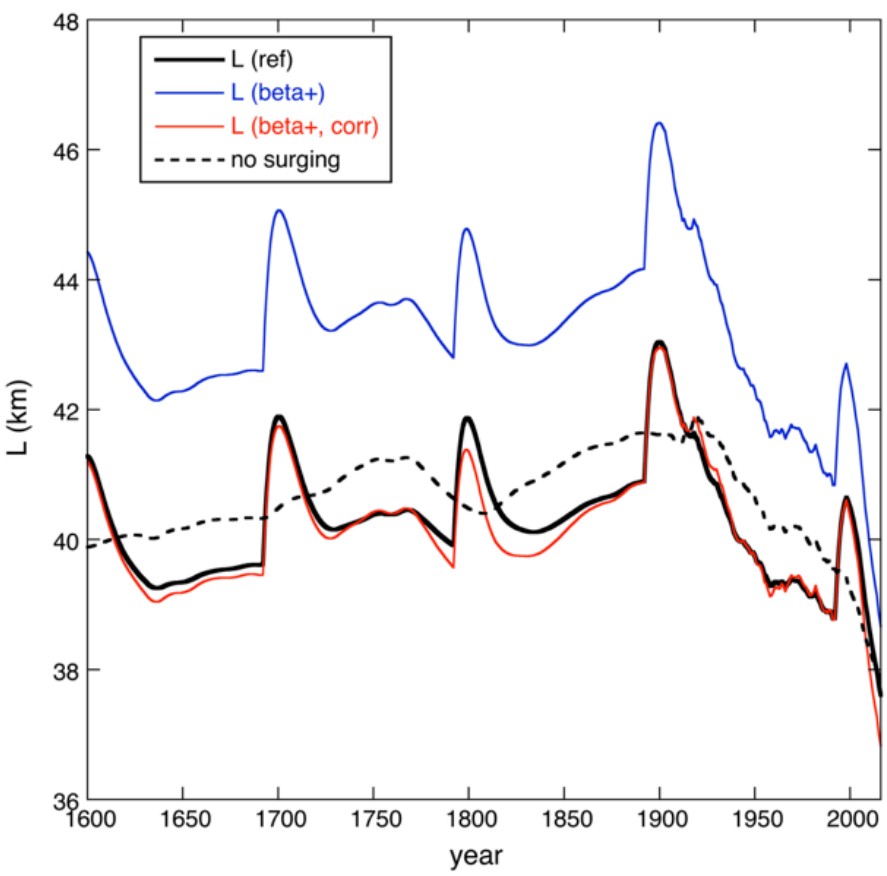


**Figure 8.** Sensitivity of glacier length to a larger balance gradient ($\beta = 0.0065$ instead of $0.0045$ m w.e. a$^{-1}$ m$^{-1}$). The reference run is shown in black, the result for the larger gradient in blue. The red curve shows the result after the model has been recalibrated by adjusting the values of $E_0$ and $E_1$. The dashed curve shows a run in which surging has been switched off (but all other parameters as for the reference run).


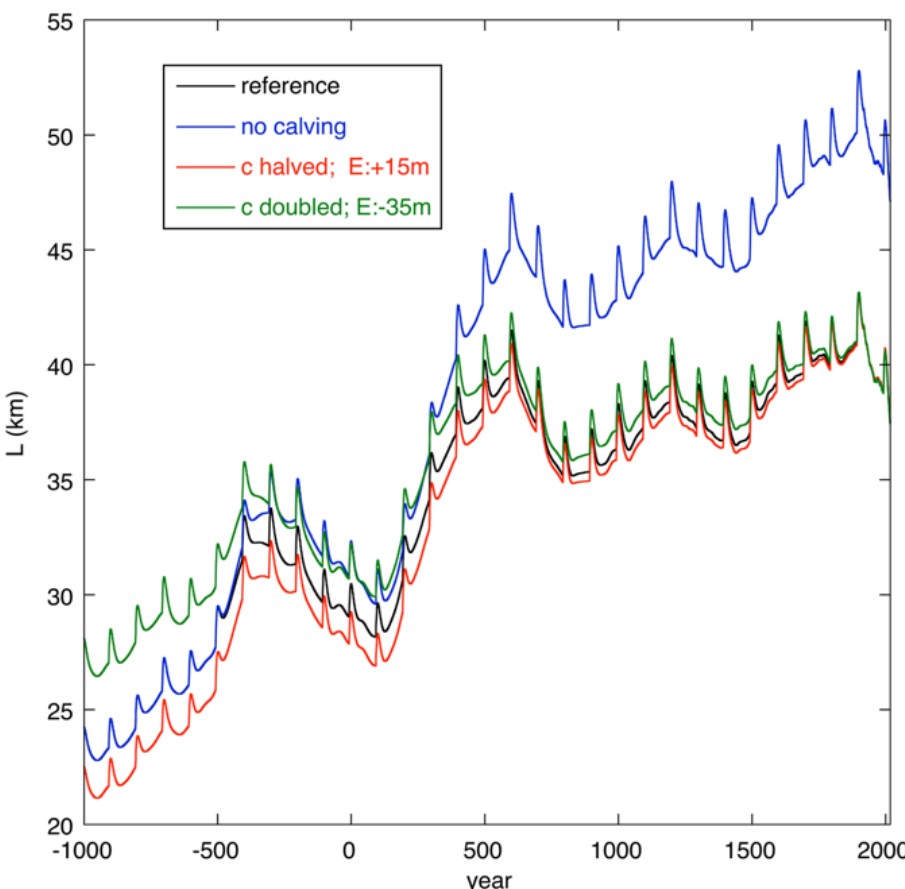

**Figure 9.** Sensitivity of glacier length to different values of the calving parameter $c$. For a halved (red curve) or doubled (green curve) value of $c$ the model has been recalibrated by adjusting the ELA (values given in legend).

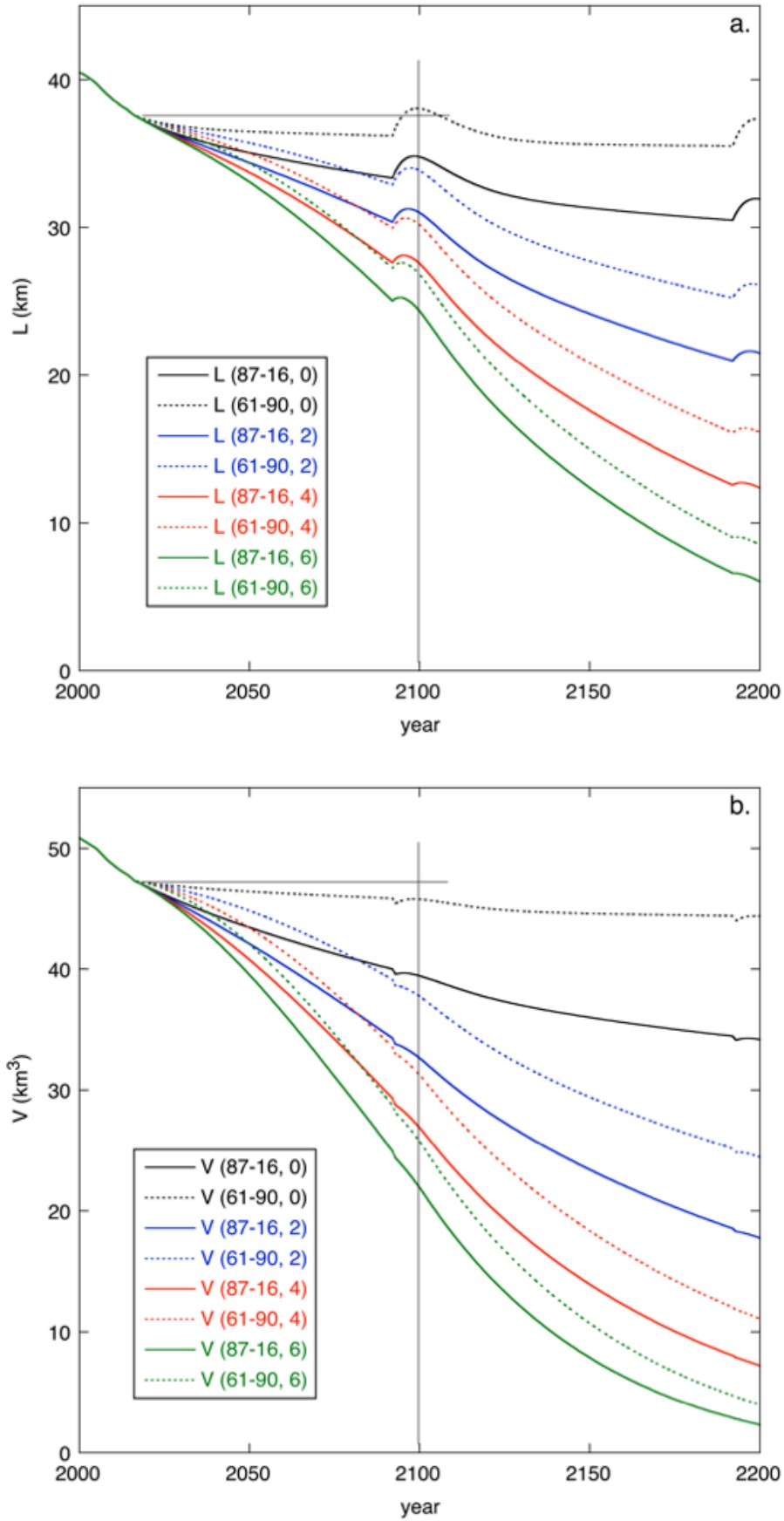


**Figure 10.** Projections of glacier length (**a**) and volume (**b**) for different climate change scenarios. The labels refer to the ELA reference period (either 1961-1990 or 1987-2016) and the rise of the ELA per year during the 21st century (in m a⁻¹).