# Peer review of "Modelling the late-Holocene and future evolution of Monacobreen, northern Spitsbergen"

_The Cryosphere, 2018_

## Referee Comment (RC1) · Anonymous Referee #1 · 21 Mar 2018

This manuscript describes a spatially-lumped model for a tidewater glacier, and then uses that model to simulate the evolution of Monacobreen in Svalbard over the Holocene and under future scenarios of climate change. The model and climate forcing are calibrated to recent observations of terminus positions of the glacier, and the impact of a recent glacier surge on glacier mass balance is also investigated. The principal novelty of the paper lies in the use of a long record of local climate during the Holocene based on lake sediment proxies of equilibrium line altitude.

Beyond the scientific content of the paper, my primary concern is that it isn't clear what the central scientific purpose of the paper is. The author has not clearly stated the

scientific goal of the paper. At the end of section 1, the author poses three questions that they will answer in the course of the paper, the first of which is: "is it possible to simulate the Holocene evolution of Monacobreen?" I'm not sure that simply answering such a question in the affirmative qualifies as an sufficient goal for a scientific paper. Clearly, it is possible to make a simulation of the Holocene evolution of this glacier, many papers have used many different kinds of numerical models to simulate glaciers evolving under forcing. But simply making a simulation doesn't make this an original scientific contribution to the literature. Other questions that one might want to answer are: "Is the simulated Holocene evolution of this glacier realistic or similar to observations?" or "What controls the evolution of this glacier during the Holocene?" These questions are grappled with to a certain extent in the paper, but that goal isn't stated clearly anywhere. Since this paper is primarily focused on describing the "minimal" model to be used, they do not explore these other questions in detail. This is my main critique of the paper: I am not clear on what we have learned from the modeling exercise as described. The author could greatly improve the paper by clearly stating what new things have been learned by doing these simulations.

Major points: 1. This is a "Minimal Glacier Model" in that it is spatially-lumped. However, there are so many processes and details included (i.e. tributary flux, SMB feedbacks, calving, surging) that I am not sure it is useful to call this model minimal. In particular, the model has so many parameterizations and associated parameters (at least 15 parameters in the main model, and another 50 parameters associated with the tributary geometry), the model will produce exactly the behavior that you have prescribed in these parameterizations. Perhaps more critically, the model has many tuned parameters (just a few: $E_0$, $E_1$, $S_0$, $t_0$, lambda, c), and it is calibrated based on just a few observations (1907, 1997 terminus positions). How does this validate the choice of those parameters to make projections? The paper could do with more discussion of how the parameters are tuned and whether those parameter values chosen represent a unique parameter set that reproduces the observed terminus positions.

As the author says, one of the benefits of minimal models is that their dynamics and processes are transparent, but it is unclear how you use this aspect of the model to add to your analysis. Some dynamical questions I had from your analysis that weren't discussed in the paper: (a) what sets the e-folding time scale of glacier response?, (b) what sets the glacier sensitivity to forcing? (c) what minimum set of processes would capture all the dynamics necessary to predict the glacier response to forcing?

2. The abstract is more of a laundry-list of model details and results, rather than a summation of the substantive takeaways from the paper. Please shorten (i.e. by removing model details). The abstract would be a good opportunity to summarize what the scientific contribution of the paper, which it does not currently include.

3. In section 4 you indicate that the glacier is more sensitive during some periods than others. Why? This bears exploring further.

4. Section 4: you have more than just the two observations of terminus position - how do these compare to the model? Do they provide unique calibration points? What observations would you need to provide stronger calibration constraints?

5. In section 5 you show that most of the change occurs after 2100. Presumably this is due to the slow e-folding time scale of the glacier. It would help to have a sense for why the e-folding time scale of the glacier in this model is so long.

6. Page 5, Line 29: you say "it is unclear to what extent the very high ELA values since 2000 represent an expression of natural variability on the decadal time scale" - however you have a long data set of ELA values that you can use to test this question. Perhaps the most novel aspect of this study is the long time series of climate forcing you have, why not use it to answer these interesting questions?

Model points: 1. What are the assumptions under which equation (3) is valid? Are their assumptions always true during transient glacier evolution (i.e. surges) in this study?

2. Write $S(t)$ and $\bar{s}(L)$ in equation 3 to make functional dependencies clear.
3. What is the physical reason that tributaries only supply mass to the main stream when they have a positive budget? Should the mass supply occur simply whenever ice flows from the tributaries to the main stream? Won't this occur even if the tributary mass balance is negative?

4. What is d in equation (9)?

5. What is kappa in equation 10? Should this equation read $H_f = H_m * max(kappa, delta)$ ? The units don't make sense as it is currently written.

6. In section 2.3 you indicate that the water depth is variable in the fjord. Should this lead us to believe that the sub-glacier topography is also variable, rather than smooth as you have assumed? How much uncertainty does lack of knowledge of sub-glacier topography introduce to the results that you present?

7. The surge function as formulated in equation 14 is an exponential function that describes a single surge, but not the periodicity of surges. The periodicity should be included in the equation itself, otherwise the reader has to figure out that $t_0$ is itself a regularly-spaced set of surge onsets, which is not so clear.

8. The fact that there is a single observation of a terminus position from 100 years before the only observed surge does not uniquely prove that the surge period is 100 years. As you say, there is no proof that a surge ended exactly in 1907 when the terminus position was observed. Even if it had, this could mean that the glacier has a surge period that is some whole fraction of 90 (1997 - 1907), like 3 surges of 30 year period or some period greater than 90. Also, it is possible that the surge period changes as the glacier retreats (e.g. because surface mass balance changes considerably).

More broadly: what is learned by including prescribed surging in this minimal glacier model? The surging is simply prescribed to exactly fit a single observation anyway. There are so many factors about the surge that are not known, I don't think you can conclude that the surface mass balance effect is small (especially since your equation

for H_m is unlikely to hold during a surge since it assumes a quasi-steady glacier profile). And if the effect of the surge is small and the focus of this study is not surging, why include this additional complication in your minimal glacier model to begin with? If you are going to keep surging in the model, the justification needs to be stronger.

Minor details:

There are typos, spelling and grammar errors throughout this manuscript. It would be helpful to the readers and reviewers if you went through the next draft more closely to correct these.

When shortening a term to an acronym (e.g. Little Ice Age - LIA), please always include the long version at the first occurrence in the paper and the acronym in parenthesis, as not all readers may be familiar with that the acronym stands for.

Page 1, Line 36: Citation needed

Page 2, Line 6: Citation needed

Page 2: it would be useful to distinguish between studies about mountain glaciers and tidewater glaciers.

Page 2, Line 20: what issue of boundary conditions? which boundary conditions? citation needed.

Page 2, Line 36: Why does surging and tributaries make modeling a challenge? Please elaborate.

Page 3, Line 6: "Freely-evolving length L"

Page 3, Line 10: why is it clear that they make a major contribution to the total mass budget.

Generally, avoid use of "it is clear" or "of course" in places where you make statements that aren't actually obvious to everyone.

[Figure]

Page 3, Line 27: In what sense was this previous study successful?

Page 4, Line 7: Citation needed

Page 4, Line 16: please explain why larger slopes mean weaker SMB feedback

Page 5, Lines 1-3: these sentences are not necessarily true (others have used complicated calving models to formulate general calving laws) and not necessary. Observations have shown calving flux varies linearly with depth at many tidewater glaciers. Just say this.

Page 5, Line 24: time scales of what?

Page 5, Line 26: I don't follow the logic of this argument for setting lambda = 15000 m. Why not just say you tuned it to give the right modern terminus position?

Page 6, Line 12: calibrated instead of measured

Section 2.5: this is not part of the model description, therefore shouldn't be included in section 2.

Page 7, Line 13: it is not necessarily the case that overdeepenings cause hysteresis.

Page 7, Line 25: explain why this observations puts a lower limit on ELA

Figure 1: what is the red filled circle on the map?

Figure 6: why not plot the observations of glacier length here for comparison

---

## Author Comment (AC1) · 23 Mar 2018

First of all, I thank the referee for the extensive comments and the many valuable suggestions to make the paper more clear and transparent. I will defintely take these into account in a revised version.

I am puzzled, however, by the slightly cynical remarks on scientific contents / questions. These remarks are a bit scattered through the document, and therefore I want to organize my initial response into four topics: (i) Scientific rationale, (ii) Why a minimal model, (iii) Calibration, (iv) The effect of surging.

[Figure]

(i) «Scientific rationale» The question of glacier response to climate warming is a central question in 'global warming science'. The issue has been tackled by different kind of models as I have described in the paper. Although simulations have been done for some of the ice caps in Svalbard (notably Austfonna), to my knowledge nobody has attempted to deal with the large complex glacier systems in Spitsbergen. The reason is obvious: the geometry is complicated, many of the glaciers are tidewater glaciers and surge as well, and overall there is very little data on bed geometry and mass balance conditions. Trying to model these glacier systems is therefore, in my judgement, a major scientific challenge. As far as I know, nobody has attempted to model the evolution of a complex glacier system like Monacobreen over a longer time span, so, frankly, I would rate my work as 'scientifically new and original'.

(ii) «Why a minimal model ?» 'Minimal' refers to the fact that the mechanics of the glacier are treated in a simple way and the model as such has no spatial resolution in the sense that numerical models have. The glacier length is the basic state variable. Adding tributaries as buckets or adding a parameterization of calving and surging does not make the model less minimal in its fundamental approach. To use a comprehensive numerical model for Monacobreen is currently not feasible. The amount of input data needed would be enormous, and the formulation of boundary conditions would involve a lot of ambiguity. Also, it is unlikely that such a model would produce surges of the right duration and amplitude, unless they are strongly imposed like in the minimal model. The reproducibility of results obtained with a minimal model is another point I want to emphasize. Anyone can code the model and have it running within a few days. I agree with the referee that some of the results, notably the e-folding response time and climate sensitivity, should be discussed in more detail in the paper.

(iii) «Calibration» Since glaciers have a memory, a projection for the future cannot be done without without a proper simulation of the past evolution. In my judgement this point has not always obtained sufficient attention. Although for Monacobreen (and most other glaciers in Spitsbergen) the information on past changes is limited, the facts can

still impose strong constraints. There is abundant geological and geomorphological evidence that Monacobreen achieved its largest size at the end of the Little Ice Age. It is not at all trivial that a model reproduces this maximum Holocene glacier stand around 1900. The case that in this paper the maximum stand can be reproduced is a significant result (apparently due to the quality of the forcing function and the suitability of the model). The correct amplitude and timing of the surge has to be included, because otherwise the starting state for the future evolution evolution would not be optimal. It is not fair to state that the minimal model has 65 parameters! Almost all of these describe the geometry and are therefore not tuning parameters. This view would imply that a numerical model with 60000 grid points would have 60000 parameters (bed elevation)!. In the model I have carefully restricted the number of tuning parameters to make it match with the information available. In fact, the two parameters describing the surge are directly taken from the observations - they are thus fixed. Except for $E_0$ and $E_1$, all other parameters ( like the calving parameter) have been taken from earlier studies (which does not imply that they have no uncertainty, but they are not used as tuning parameters). With respect to the bed profile: the values of lambda has not been tuned to give the correct front position. However, slight variations in this parameter were considered as described in the discussion. The bathymetry in the fjord shows irregularities (mainly moraines), but it unlikely that such irregularities extend underneath Monacobreen because its surface is so smooth. The fact that the model reproduces the maximum stand around 1900 as well as the correct rate of retreat after the last surge is a major achievement and not directly forced by the calibration. I regret that the referee does not acknowledge this is any way.

(iv) «The effect of surging» The referee states "I don't think you can conclude that the surface mass balance effect is small (especially since your equation for $H_m$ is unlikely to hold during a surge since it assumes a quasi-steady glacier profile)". Here I do not agree. First of all, the relation between the mean ice thickness and glacier length does not assume a steady-state profile. However, more importantly, the mass balance perturbation depends only on the change in the MEAN surface elevation, not

on its spatial distribution (because the balance rate is a linear function of altitude). So when the change in glacier length is known/prescribed from observations, the change in mean surface elevation automatically follows from mass conservation. This is implicit in the model formulation and in my view an elegant way of dealing with the effect of surges (more explicitly discussed in my paper on Abrahamsenbreen (Oerlemans et al., 2015)). So I think that the conclusion about a minor effect of surging on the long-term evolution of Monacobreen is firm (and in fact a consequence of the small surge amplitude: the change in length is only about 5 % of the total glacier length).

Reference: J Oerlemana and W J J van Pelt: A model study of Abrahamsenbreen, a surging glacier in northern Spitsbergen. The Cryosphere, vol 9, 767-779 (2015).
* * *

---

## Referee Comment (RC2) · Anonymous Referee #1 · 26 Mar 2018

Thank you for your quick response. I think if you clearly state (in the paper) the purposes that you have outlined in this initial response, the paper will be greatly improved and the reader will have an easier time of discerning the most important points of the paper. For example, if there were a sentence at the end of section one that read (something like): The central purposes of this study are to: (1) . . .

If I understand correctly, you have identified several purposes of the paper, all of which I find to be valid, among them:

(a) The modeling of a large complex glacier system in Spitsbergen and its response to future climate change, which is of interest to those seeking to understand future sea

level rise from glacier melting.

(b) The development of a simple model to simulate a relatively complex system that could also potentially be used to model other complex glacier systems.

I don't disagree with your other points, and I think if the text is clarified to explain these arguments (e.g. what assumptions do go into deriving this mean ice thickness; why minimal model results are more reproducible than those for a more complex model in the absence of well-constrained observations) and those I outline in my initial review, then the average reader will be able to understand and reproduce your results more easily.

---

## Referee Comment (RC3) · Anonymous Referee #2 · 19 Apr 2018

Summary

The manuscript uses a simplified glacier model to estimate the evolution of Mona-cobreen, a calving glacier in northern Spitsbergen. The model includes surging and calving parameterizations but neglects spatially resolved ice dynamics and asserts a simplified bed topography. Mass balance and calving parameters are asserted from previous studies in Svalbard. The main stream of Monacobreen is fed by a number of side glaciers, 10 of which are modelled by assuming simplified geometries. The model response time is explored with step changes in the ELA. The model is forced with an estimated past-ELA record derived from lake sediments and the conversion

of future air temperature estimates into ELAs. The mid-Holocene to 2099 CE length evolution of Monacobreen is explored. The sensitivity of the model to calving, surging, and the presence of the largest tributaries is explored. The modelling framework is supported by a number of papers and books by the author. There are a few typos in the manuscript and it is generally well written, though there are some portions of the text where more explanation is needed.

This manuscript is novel in that it seeks to address the difficult problem of modelling large tidewater glaciers, with numerous branches, and no mass balance, ice thickness, or velocity data. But when little data is available to constrain parameters, uncertainties and the number of assumptions grow. I think the approach is interesting and useful but because of the large number of assumptions and asserted parameters (based on very limited local data) I find it hard to believe that the results actually represent the evolution of Monacobreen. In my mind this work is an exploration of the Minimal Glacier Model with the best possible parameters asserted in a remote region. It follows that some of the inferences are overstated based on the analyses performed in the manuscript. I think the overall motivation for the study needs to be improved and refined. And the model explanation needs to be much more clear about the assumptions made in the derivation of the model. The way the manuscript is written it is hard for me to tell what those assumptions were at first. I had to look a several other papers to get a sense for the assumptions. I think some of the model variables and parameters need further explanation. Where possible quantitative sensitivity analyses should be supplied to the reader instead of assertions that certain the model is not sensitive to certain choices. I would like the sensitivity and parameter analysis to be greatly expanded. The evaluation should be quantitative and relevant specifically to this study, and should also isolate single parameters. The low computational cost of the model means these analyses can be done rapidly. A table with the full list of parameters (asserted and tuned) would be vary helpful. The effect of changing the parameter on length say from a reference state would also be helpful. For example, how sensitive is the model to changes to the asserted mass balance gradient? The term is essential to the model

sensitivity through the B_tot term in equation 4 but there are no measurements from the actual glacier. The discussion should be expanded to highlight the implications of the assumptions made in the modelling approach. Items that should be discussed in more detail: 1) the neglect of bed topography/variability on calving rates; 2) the sensitivity of the model to changes in the mass balance gradient (which controls the sensitivity of the glacier to changes in the ELA (and seems to be most important climate parameter)); 4) quantified sensitivity of the model to the asserted bed profile; and 5) quantified uncertainty to the estimated side basin geometries (the geometry is simplified so there must be significant uncertainty).

A couple specific items of concern:

Calving glacier length change is strongly controlled by bathymetry and bed geometry. I point to Vieli et al., 2001 (from the abstract):

"Length changes of tidewater glaciers, i.e. especially rapid changes, are dominantly controlled by the bed topography and are to a minor degree a direct reaction to a mass-balance change. Thus, accurate information on the near-terminus bed topography is required for reliable prediction of the terminus changes due to climate changes."

Vieli et al., 2001 show that small bed fluctuations on order of 100 m can pin the calving front where the bed shallows. You note that there are known fluctuations on order of 100m in the bathymetry in front of the modern calving front. It is a major assumption that the bed monotonically declines. This should be discussed in the paper and is a major caveat to the current approach. The lack of bed data makes me skeptical of the length fluctuations outside of the historical record.

I am also concerned about the Holocene temperature history that is implied by the ELA sensitivities (35 m/ K and $-2.25$ m /% (page 8 line 4)) and lake-derived ELA record. If we assume that the ELA change ($\sim$250 m; Figure 6) was accommodated only by temperature changes then 4 thousand years before 1900 the air temperature would have been 7.1 degrees warmer. The manuscript points to temperature as being the

primary control of the Holocene ELA decline: "Due to the declining summer isolation, at the end of the Holocene Climatic Optimum the Arctic climate cools and the equilibrium line drops." (page 8 line 16-7). If we assume that half of the ELA change is accommodated by changes in temperature then the air temperature perturbation at the start of the ELA forcing would be 3.6 degrees and precipitation would have had to have been 56% lower. This implies that precipitation would have had to decrease when air temperatures were warmer. This is opposite of the often assumed increase in precipitation with temperature as warm air holds more moisture. These climate scenarios are extreme and highly unlikely (Kaufman et al., 2004 Holocene thermal maximum in the western Arctic (0-180 deg. W). This is a concerning because the future climate scenarios are based on these low dELA/dT numbers which in turn imply outlandish Holocene air temperatures.

Detailed comments:

Abstract: I think the abstract could be revised to include a better explanation of the motivation for this study. It would also benefit from a synthesis of the model results currently presented in the abstract.

Page 2:

Line 19-21: A citation would be helpful here as I am not sure which boundary conditions you refer to.

Page 3:

Section 2 Glacier model: This model description does not present the model clearly and it does not explain what the model assumptions are. Does the model neglect ice dynamics? Does the model assume equilibrium to determine glacier length? How does the model represent its response time? Is ice thickness resolved throughout the domain? These questions should all be answered here. They are fundamental to assessing the viability of the model to this specific application. Oerlemans, 2011 is referenced but I am not sure which equations in the book are actually relevant to the form of the Minimal Glacier Model used in this manuscript. Overall, the manuscript assumes too much knowledge of previous papers published on Minimal Glacier Models.

Some of the basic conceptual framework for this manuscript is outlined in the discussion but it needs to be in the modelling section.

Line 5-11: I think it should be mentioned out right how the model treats ice dynamics and that you assume an idealized bed profile.

Page 4:

Line 10: Mass balance gradients vary considerably over short spatial scales depending on local precipitation and air temperature lapse rates

Line 21-24: How sensitive are your results to uncertainty in these geometric parameters?

Page 5:

Line 7: what is kappa?

Line 18: main stream?

Line 20: "Until today" It seems this should be revised as it is a bit confusing what you mean.

Line 22-23: The reason for this is not clear. Please explain or add a citation.

Page 6:

Line 8-9: It would be helpful to see the actual, known bathymetry along side the idealized version in figure 3.

Line 10-11: "The mean ice thickness for the present state of the glacier is about 300 m." Is this for the actual glacier or in the model? Figure 3 shows that ice thickness is substantially smaller than 300m when the glacier surface is connected to the bed

profile used in the model. What is going on here?

Page 7:

Section 2.5 How is the response time determined?

Line 7-8: What is the justification for perturbing the ELA of basins 1-3? These are additional parameters that you are asserting without any local constraint or optimization.

Line 11: I think it is more fair to represent that it turns out the climate sensitivity of the model is large as opposed to the glacier.

Line 14: What equation is the e-folding response time based on?

Page 8:

Line 3-4: These asserted ELA sensitivities are vital to your modelling yet you do not explore the sensitivity of the model to ELA sensitivity or the mass balance gradient. van Pelt et al., 2012 should be cited for the ELA sensitivities as that is were the numbers come from originally.

Line 38: check spelling.

Page 10:

Line 27: check spelling.

Line 30: How similar are the results? It would be beneficial to the reader if you provide a quantitative evaluation here. Right now your assessment of the model uncertainty is not fully fleshed out as I would like to assess the degree to which the bed topography effects the resulting model evolution.

Line 31-32: You can quantify the uncertainty of the model parameters and tell us exactly how important the forcing and the parameters are though. As it stands now you are asserting that they have less uncertainty but not showing that it is true.

Page 11:

Line 1: Extra period.

Figures: Some figures have panels labelled as a and b others do not.

Figure 1. Which of these length constraints are used to verify the model? It is not clear from the figure.

Figure 2. It would be helpful if you delineated the side basins as you represent them with the parameters in table 1.

Figure 3. Please also include the modelled ice thickness in this figure for the same time period as the glacier surface elevation. Or clearly explanation at the start of the paper that this model does not spatially resolve ice thickness.

Figure 5. Please explain what the lines in the legend represent in the figure caption.

Figure 6. It would be helpful if you put the model constraints in the lower panel of this figure. Should be labelled as 'yr CE'

Figure 8. This is a perfect figure to incorporate a sensitivity of the model to changes in as many parameters as possible, $\alpha$, v, the mass balance gradient, and assumed surge parameters (period, magnitude) as well as to changes in the guessed bed topography.

Figure 9. The solid lines and dots need to be better explained in the caption. I am confused as to why the reference period is an important issue to explore here.

---

## Author Comment (AC2) · 24 Apr 2018

I am very grateful to the referee for his/her thorough review, which acknowledges the originality of the approach taken to model a complex tidewater glacier system, but at the same time has significant criticism on, notably, (i) the model description (especially concerning the choice of model parameters and bed topography), (ii) the documentation on parameter sensitivity (too limited), (iii) response time and sensitivity, and (iv) the meteorological interpretation of the imposed ELA-history and future scenarios. I plan to revise the paper thoroughly on these point (apart from accommodating many small useful suggestions).

[Figure]

(i) Description of the Minimal Glacier Model (MGM) Although the MGM has been described in a number of publications where all the relevant equations can be found, the referees request a more extensive description with a better explanation of the basic assumptions. This will be accomodated in the revised version. Notably, a text along the following line will be included: In the MGM there is no explicit resolution in the sense that the ice thickness is calculated along a flowline. The mean ice thickness is related to glacier length and mean bed slope. It be noted that the mean bed slope changes when the glacier length changes. The relation used is based on extensive experimentation with a numerical flowline model (Oerlemans, 2001). The parameter alfa in fact is a measure of the bed resistance to the glacier flow. For glaciers on Svalbard this resistance is very low and the values of alfa are correspondiongly small (as compared for instance to glaciers in the Alps). The use of the relation between length, slope and thickness implies that the height-mass balance feedback is included in the model. In fact, as has been demonstrated in Oerlemans (2011, Figure 5.8), the model even fairly accurately reproduces the hysteresis implied by an overdeepening. When the balance profile with height is linear, only the mean ice thickness enters the expression for the surface mass budget. So the fact that the ice thickness is not calculated as a spatial variable has no effect on the calculated climate-driven evolution of the glacier. This does not apply to the parameterization of calving, however. In the MGM the glacier front is always vertical and a fraction (kappa) of the mean ice thickness (or equal to the floatation thickness when this would be larger). Although this appears to be a rigorous approach, it allows a smooth transition from a land-based to calving glacier and vice versa [note: to my knowledge a full cycle of a land-based glacier into a calving glacier, and the way back, has never been simulated with an 'advanced' numerical model]. The method has been applied succesfully to Hansbreen (Oerlemans et al., 2005). Referee 2 notes that variations in the bed topography may have large effects on the calving rate and thereby on the position of the glacier front. According to the measured bathymetry in the Liefdefjorden, these variations with an amplitude of 10 - 50 m (not 100 m) are irregularly spaced and consist mainly of deposited moraines. It is unlikely that a similar

bed would currently be present under Monacobreen with its very smooth surface. It is therefore not relevant to include a map of the bathymetry of the Liefdefjorden in the paper. I also don't think that the details of a bed profile matter for the glacier evolution on longer time scales, unless there are very marked jumps in bed or side geometry that could serve a pinning points. However, this does not seem to be the case.

(ii) Parameter values. To provide a better overview of the parameters used in the model a table will be included, in which references, when appropriate, will also be listed. New parameters for the bed profile will be determined by requiring consistency between ice thickness and observed surface profile, for the value of a taken from Kronebreen (which is based on observations; Hagen and Saetrang, 1991). Referee 2 mentions specifically that the effect of the balance gradient could be important and should be studied / presented. It has been demonstrated in many studies (with MGMs as well as numerical glacier models) that the effect of the balance gradient on the equilibrium length is small. At the same time the effect on the response time is significant. I will explain this in more detail in the revised version. The more extensive parameter sensitivity study requested by Referee 2 will imply that one more figure will be added to the paper. I suppose that is no problem.

(iii) Response time. The response time is an e-folding response time. It is calculated from the response curve in a straightforward way.This will be explained better in the revised paper. The large value (350 years) is mainly due to the very small slope of the glacier and the associated strong height-mass balance feedback, which is known to lead to larger response times (e.g. Oerlemans, 2001).

(iv) Climate forcing. It is clear that here a more extensive discussion is needed. Referee 2 calculates that, in order to explain the high ELA values 4000 YBP, a 7 C higher temperature would have prevailed. He/she refers to Kaufmal et al. (2004), who suggest that the Holocene thermal maximum would have been +1.5 K. Several remarks can be made concerning this apparent discrepancy. Later paleoclimatic studies have show the Kaufman et al. estimate is really too conservative (refs will be included). With respect

to the apparent very high value of the ELA in the mid-Holocene: summer insolation was typically 30 W/mˆ2 higher than today, which no doubt had a strong effect on the summer melt rates and thereby on the ELA (apart from the temperature effect). So it is not necessary to attribute the entire rise in ELA from a higher temperature alone. In any case, it is the ELA variations that drive the glacier evolution, and the ELA reconstruction used seems to be very robust. With regard to the future projections: I think that the use of two different reference periods is useful and points to an aspect that is often not getting enough attention: the strong dependence of projections on uncertainty in the initial state. I am not quite sure why Referee 2 finds this unclear, but I will try to come up with a better explanation.

References:

Hagen J. O., Saetrang A.: Radio-echo soundings of sub-polar glaciers with low-frequency radar, Polar Res., 9 (1), 99-107, doi: 10.3402/polar.v9i1.6782, 1991.

Kaufman et. Al: Holocene thermal maximum in the western arctic. Quat. Science Rev. 23 (5), 10.1016/j.quascirev.2003.09.007 , 2004

Oerlemans, J.: Minimal Glacier Models. Second edition. Igitur, Utrecht University, ISBN 978-90-6701-022-1, 2011.

Oerlemans, J:ÂăGlaciers and Climate Change. A.A. Balkema Publishers, 148 pp. ISBN 9026518137, 2001

---

## Author Response (AR1)

**Documentation of the revision**

PLEASE NOTE:

• Major changes are listed here below – in addition many small alterations / corrections / additions were made following the suggestions of the referees.

• In the revised manuscript new parts or substantially changed parts are *shaded*. I tried to work with the text editor, but it became very messy.

• Figures have been changed and one new figure has been added (sensitivity for the calving parameter *c*).

• I tried to draw basin outlines in Fig. 2 as suggested by one of the referees. However, the result really disturbs the satellite image and it adds no new information (basin characteristics given in Table 2). So in the end I did not change the picture.

• New references have been added.

• I now use the BCE / CE system to indicate years in the text.

**Explaining in more detail the significance of the study (lines 83-94 were added / revised ):**

Monacobreen, with a distinct calving main stream and a large number of tributaries, represents a glacier system that is typical for Spitsbergen. The complexity of the geometry as well as the limited amount of data make it a real challenge for a modelling study. Nevertheless, the question of how the mass of such glacier systems will change in the near future has to be considered, and the approach taken in this study is an attempt to do this in a meaningful way. A MGM provides a reasonable match between the paucity of data and an integrated mass budget approach, in which glacier mechanics are parameterized in a simple way. The larger glacier systems on Svalbard presumably have long response times. The strategy of using observations on former glacier stands for calibration before integrating the model into the future is tested in this paper. It is envisaged that the methodology can be applied to other complex glacier systems in Svalbard and the Arctic.

With respect to Monacobreen, the following more specific questions will be addressed: (i) Is it possible to simulate the broad characteristics of the late Holocene evolution of Monacobreen ?; (ii) To what extent does regular surging effect the mass budget and long-term evolution of the glacier ? ; and (iii) What is the likely range of mass loss in the coming centuries for different scenarios of climate change ?

**A further explanation on the interaction between geometry and ice thickness in the MGM, as well as a few lines on the method of solution (lines 130-138 were added ):**

[revised manuscript text omitted]

**A more extensive explanation of why the ELA is set to lower values for basins 1, 2 and 3:**

This is done to take the decline of the ELA as mapped in Hagen et al. (1993) into account. In fact, without the lowering of the ELA the net mass budget of basin 1 (Seligerbreen) would be negative and the tributary could never supply mass to Monacobreen (as it did until recently). Altogether, the ELA map of Hagen et al. (1993) appears to be consistent with the mass budgets of the tributaries.

**A further comment on the sensitivity of the ELA to temperature and precipitation anomalies, explaining why the temperature sensitivity is relatively low on Spitsbergen (incl. a new reference), as well as a discussion on temperatures during the Holocene Climati Optimum:**

The relation between the ELA and temperature / precipitation is based on calculations with an energy balance model, as described in Van Pelt et al. (2012) and Oerlemans and Van Pelt (2015). The sensitivities are $\partial E/\partial T = 35$ m K$^{-1}$ and $\partial E/\partial P = -2.25\ m\ \%^{-1}$, where $T$ and $P$ are perturbations of the annual mean temperature and precipitation. It should be noted that the value of $\partial E/\partial T$ is rather small compared to values found for glaciers in a midlatitude alpine setting, which are of the order of 100 m K$^{-1}$. This stems from the fact that summer temperature anomalies over Spitsbergen (and in general over the Arctic region) are much smaller than mean annual temperature asnomalies. Since summer temperature determines to a large extent the ELA perturbation, the net effect is that the sensitivity to an *annual* temperature anomaly is relatively small (for a further discussion on this point, see Van Pelt et al. (2012).

As shown in Fig. 6a, the variation of reconstructed ELA values from mid-Holocene times until today have a typical range of 200 m. If this would solely be a temperature effect the drop in ELA since the mid-Holocene would correspond to a 5.7 K decrease in temperature (according to the sensitivity referred to above). This is more than reconstructions of mid-Holocene warmth in the Arctic actually suggest, which are in the 2 to 4 K range (e.g. CAPE, 2006; Bradley, 2016; Axford et al., 2017). However, there is also a direct effect of changes in orbitally-driven insolation variations. The differences in summer insolation between mid-Holocene and present day are between 5 and 10%, depending on the precise location and definition of the summer season (Berger and Loutre, 1991). The increased insolation certainly caused higher melt rates in the mid-Holocene, and thereby a higher equilibrium line.

**A comment on the time scale (section 2.5):**

With the adjusted value of $\alpha$, the e-folding response time is about 250 years. A short discussion has been added to put this value in perspective (including some additional references):

The time scale of about 250 years can be compared with an estimate of the much used volume time scale $\tau_J$ defined by Jóhannesson et al. (1998):

$$\tau_J = -H^*/\dot{b}_{x=L}\ , \tag{15}$$

where $H^*$ is the maximum ice thickness and $\dot{b}_{x=L}$ is the balance rate at the glacier front. Using 350 m as a maximum ice thickness and $-2.5$ m a$^{-1}$ as a typical balance rate yields a value of about 140 a. However, as demonstrated by Oerlemans (2001) and confirmed by Leysinger Vieli and Gudmundsson (2004) with a higher order numerical glacier model, for glaciers with small slopes the altitude-mass balance feedback makes the time scale considerably longer. Since Monacobreen has a very small average slope (0.027), the value of ~ 250 a is a plausible one.

**Fig. 6c (close-up of simulated glacier length):**

The observed glacier stands are now shown in Fig. 6b as blue dots, illustrating that the calibration with the parameters $E_0, E_1, S_0, t_s$ works. A sentence is added to make clear that the calibration procedure is transparent:

At this point it should be recalled that the tuning procedure is straightforward: four calibration parameters $\{E_0, E_1, S_0, t_s\}$ have been used to match: (i) the LIA maximum stand, (ii) the glacier stand at the onset of the surge, (iii) the amplitude of the surge, and (iv) the time scale of the surge.

[Figure]

**Special section on sensitivity tests**

Section 4 (Holocene evolution of Monacobreen) has been split up into two subsctions to provide a more elaborate treatment of the sensitivity to some model parameters (balance gradient, surge amplitude, calving parameter). This also involves an additional figure. The text has been adjusted accordingly, in short:

The balance gradient $\beta$ has been varies across a wide range, and for one particular value the effect on the simulation is shown (in Fig. 8). Since the larger value of $\beta$ implies a larger glacier, the model has been recalibrated by adjusting the ELA (also shown in the figure). The conclusion is that the choice of $\beta$ is not crucial for the simulation, and that the Holocene evolution does not change in a qualitative sense.

The effect of the surge amplitude has also been investigated as described in the text. In the figure the result of a run without surghing is shown.

Fig. 9 (new figure, see below) focuses on the role of the calving parameter $c$. Cases of no calving, calving parameter halved and calving parameter doubled are compared. In each case recalibration has been done to match the observed glacier length record. The effect of changing $c$ on the long-term evolution of Monacobreen appears to be modest.

[Figure]

**Main changes in section 5 (Future evolution of Monacobreen).**

An additional remark on the relation between changes in the ELA and meteorological variables.

A more extreme scenario has been added, e.g. one in which the ELA increases by 6 m/yr.

The choice of reference period: apparently the explanation was not quite adequate, and the relevant paragraph has been changed into:

> When making projections future climate change scenarios the outcome depends on the choice of the reference period. Starting from a warm year (e.g. 2015, ELA = 809 m) and increasing the ELA by a certain amount will give a very different result from starting in a cold year (e.g. 2014, ELA 668 m). Therefore the reference ELA should be a mean value over a longer period. Moreover, it is unclear to what extent the very high ELA values since 2000 represent an expression of natural variability on the decadal time scale, or are a direct response to greenhouse-gas induced warming. To deal with this uncertainty, two 30-yr reference periods were used to define the ELA perturbation associated with the projected climate change: (i) 1987-2016, i.e. the most recent 30-yr period; and (ii) 1961-1990 as the last official period to define the climatology. The resulting eight projections of glacier length are shown in Fig. 10a. The integrations are extended until 2200, and the ELA-perturbation is kept fixed after 2100. The curves immediately make clear that typically half of the response to 21st century warming will come after 2100.

**Main changes in section 6 (Discussion).**

An additional remark on the conclusion that surging is relatively unimportant for the long-term evolution of Monacobreen:

> It should be noted that the perturbation of the mass budget is solely determined by the redistribution of mass, not by the details of how this distribution actually takes place. When a glacier increases its length during a surge, the change in mean surface elevation is entire dictated by the conservation of mass, not by the details of the surging mechanism. This implies that conclusions about the effect of surging on the long-term mass budget can be drawn even when the surging process is not fully understood.

> *Note: Referee 1 actually suggests to leave out the surging mechanism. On this point I really disagree. It is simple to include it (just a few lines of code), it makes the tuning more straighforward (how should I do this without the surge?), and it does not violate any conservation law or physical principles. I think I use an elegant way to study the effect of surging. There is no solid argument to leave it out.*

A discussion on the role of calving:
The experiments with different values of the calving parameter are discussed, and the effect of small-scale variations in the bed toppography is put into perspective:

> Calving has a significant effect on the total mass budget of Monacobreen, but different values of the calving parameter do not change the qualitative evolution of the glacier during the Holocene very much. The range over which Monacobreen fluctuates is somewhat smaller for a larger calving parameter (the green curve in Fig. 9). This is understandable for a bed profile that slopes downward along the flowline, because the front of a growing glacier comes into deeper water and the mass loss by calving increases.
> It has been observed that on shorter time scales details of the bathymetry may have significant effects on the calving rate and thereby on the position of the glacier front (e.g. Vieli et al., 2002). According to the measured bathymetry in the Liefdefjorden, these variations with an amplitude of 10 - 50 m are irregularly spaced and consist mainly of deposited moraines. It is unlikely that a similar bed would currently be present under Monacobreen with its very smooth surface, or existed in the fjord before the glacier started to advance in late Holocene times. Therefore it does not seem meaningful to include a map of the present-day bathymetry of the Liefdefjorden in one way or another. Probably, the smaller features of the bed profile do not matter too much for the glacier evolution on longer time scales, unless there are very marked jumps in bed or side geometry that could serve a pinning points. However, this does not seem to be the case.

**References added:**

Axford, Y., Levy, L. B., Kelly, M. A., Francis, D. R., Hall, B. L., Langdon, P.G., and Lowell, T. V.: Timing and magnitude of early to middle Holocene warming in East Greenland inferred from chironmids, Boreas, doi.org/10.1111/bor.12247, 2017.

Berger, A., and Loutre, M. F.: Insolation values for the climate of the last 10 million years. Quat. Sci. Rev., 10, 297-317, 1991.

Bradley, R. S.: Holocene climate change in Arctic Canada and Greenland, Qua. Sci. Rev., doi.org/10.1016/j.quascirev.2016.02.010, 2006.

CAPE-Last Interglacial Project Members: Last Interglacial Arctic warmth confirms polar amplification of climate change. Quat. Sci. Rev., 25, 1383-1400, doi.org/10.1016/j.quascirev.2006.01.033, 2006.

Leysinger Vieli, G. J.-M. C., and Gudmundsson, G. H.: On estimating length fluctuations of glaciers caused by changes in climatic forcing. Journal of Geophysical Research 109, F01007, doi:10.1029/2003JF000027, 2004.

Martín-Moreno, A., Alvarez, F. A. and Hagen J. O.: 'Little Ice Age' glacier extent and subsequent retreat in Svalbard archipelago, The Holocene, 27 (9), 1379-1390, doi.or/10.1177/0959683617693904, 2017.

Todd, J., and Christoffersen, P.: Are seasonal calving dynamics forced by buttressing from ice melange or undercutting by melting? Outcomes from full-Stokes simulations of Store Glacier, West Greenland. The Cryosphere, 8, 2353-2365, doi: 10.5194/tc-8-2353-2014, 2014.

Van Pelt, W. J. J., Oerlemans, J., Reijmer, C. H., Pohjola, V. A., Petterson, R., and Van Angelen, J. H.: Simulating melt, runoff and refreezing on Nordenskiöldbreen, Svalbard, using a coupled snow and energy balance model. The Cryosphere 6, 347-360, doi: 10.3189/2012/JoG11J217, 2012.

Vieli A., Jania, J., and Kolondra, L.: The retreat of a tidewater glacier: observations and model calculations on Hansbreen, Svalbard, J. Glaciol., 48 (163), 592-600, 2002.

---

## Author Response (AR2)

RESPONSE (shaded) to second review by Referee 1 (points from first review that were not addressed).

*First of all:*
*I went carefully through the manuscript again in surge for errors. Last time this was done in a bit of a haste (the referee was right at this point) due to the deadline. I apologize for that.*

2. Write S(t) and \bar{s}(L) in equation 3 to make functional dependencies clear.
I have now done this. However, it remains a bit ambiguous: H_m and L are also functions of t, of course. But ok, no problem to do it in the way suggested by the referee.

3. What is the physical reason that tributaries only supply mass to the main stream when they have a positive budget? Should the mass supply occur simply whenever ice flows from the tributaries to the main stream? Won't this occur even if the tributary mass balance is negative?
If a tributary has a negative mass budget it will shrink and become detached from the main stream. Since it is assumed that the tributaries are in equilibrium with the prevailing value of $E$, this implies that they cannot deliver mass to the main stream. I have clarified this better in the text.

5. Equation 10: Should this equation read H_f = H_m * max(kappa, delta) ? The units don't make sense as it is currently written.
Water depth is missing - sorry about that. It should be H_f = H_m * max(kappa, delta * d)

7. The surge function as formulated in equation 14 is an exponential function that describes a single surge, but not the periodicity of surges. The periodicity should be included in the equation itself, otherwise the reader has to figure out that t_0 is itself a regularly-spaced set of surge onsets, which is not so clear.
I looked at this, trying to go to a formal mathematical notation. This becomes awkward and confusing, however (but I am open for a suggestion...). Instead I have added t larger or equal than t_0 in the equation, introduced the symbol T_s for the duration of the surge cycle, and explained that the values of t_0 are regularly spaced at intervals T_s.

8. The fact that there is a single observation of a terminus position from 100 years before the only observed surge does not uniquely prove that the surge period is 100 years. As you say, there is no proof that a surge ended exactly in 1907 when the terminus position was observed. Even if it had, this could mean that the glacier has a surge period that is some whole fraction of 90 (1997 - 1907), like 3 surges of 30 year period or some period greater than 90.
Well, this is a bit of a theoretical remark as there is absolutely no field evidence that a surge occurred in between.
Also, it is possible that the surge period changes as the glacier retreats (e.g. because surface mass balance changes considerably).
Yes, of course.....

Section 2.5: this is not part of the model description, therefore shouldn't be included in section 2.
I considered this section to be a basic test of the model before going to the climate applications, but ok, I changed it into a section of its own (section 3 now).

-It is not necessarily the case that overdeepenings cause hysteresis.
Well, I don't agree. I have done experiments with MGMs, SIA-models, Elmer, and it seems that an overdeepening ALWAYS leads to hysteresis (when the balance rate depends on altitude). I would challenge the referee to come up with a run with a glacier model NOT showing hysteresis. But, ok, it is not so important and I have added the word *probably*.

Line 298: explain why this observations puts a lower limit on ELA

Well, if there is no glacial activity the equilibrium line must be above the highest point of the mountain ridge. I thought this is obvious. Anyway, I have added: *(in this case the ELA has to be above the mountain top)*.

Other Major points:
-It would perhaps be helpful to do a close revision (or have another person do a close read) of the manuscript for grammatical and spelling errors.
I have done this.
There are many places where punctuation (commas mostly) are needed to separate off transitional statements at the beginning of sentences. Also, new paragraphs often seem to start in odd places. I have attempted to find all the spelling errors, but these should have been fixed before review.
I made some improvements...

-I still find the abstract to be too long and including too many details about the study. Perhaps you can pick the 3-5 most important results of the study and condense the abstract to highlight those. Otherwise, it is difficult for the reader to determine the most importance results from the abstract.
I rewrote the abstract according to the Editor's instructions. Hopefully it is better now. By the way, in the orginal abstract I included the reference to Røthe et al. (2014), solely because I did not want to make the impression that I produced the reconstruction myself !

-Line 99: given that "flowband" is typically used to refer to models that resolve thickness and velocity horizontally along the glacier flow, this statement is misleading. I would suggest using the term "lumped" or "box" to refer to the model, which has a single prognostic variable for the entire distance downstream.
A bit of a picky statement. The dependence of the bed height on x plays an important role in the dynamics. But ok, I changed 'modelled as' into 'considered to be'.

Other Minor points:
Lines 38-44: The first paragraph could be condensed into two sentences.
I don't agree.

Line 50: remove "very much"
This point has been underestimated by many (most) modellers, and here I want to make a staement; I changed 'very much' into 'strongly'.

Line 55: be more specific than "entire glacier population"
Added 'on the globe'.

Line 56: A different line from what? Be more specific at the beginning of a new paragraph.
Somewhat picky comment: different from the models mentioned in the section before, of course.

Line 63: "Three-dimensional"
OK.

Line 64: "the issue of boundary conditions in steep terrain has not yet been resolved in a satisfactory way" - be more specific
I have added: *(dynamic simulation of steep ice-free slopes, mass conservation)*

Line 71: MGMs require few computational resources
Eh... this suggestion is poor English. I think the sentence in the manuscript is fine.

Line 76: In this paper, a MGM...
OK.

Line 79, 87: substitute "difficult" for "a real challenge"
I did that in line 79, not in 87.

Line 91: "The larger glacier systems on Svalbard presumably have long response times" - provide a citation or some reasoning behind this statement
I added: *, because they have small slopes and are subject to relatively dry conditions*

Line 103: delete "it is clear that"
OK.

Line 120: S is the "surge function" (described at length in section XX), then remove "The form of S(t) will be described later." on line 123
OK.

Line 123: Cite Oerlemans 2011 here
OK.

Line 125: delete "apparently"
OK.

Line 145: Euler scheme of what order? (I assume first - state this)
OK.

Line 145: Tests
OK.

Line 146: replace "a" with "year" throughout the paper
This is a never ending story, it seems... I checked with the official rules and, really, 'a' is the official and only symbol for time unit of a year. Not 'y' or 'yr' or 'year' or 'yrs'. So I think that also the glaciological community should try to become more consistent here !

Line 151: cite needed
OK.

Line 155: Equilibrium line altitude is used as climate forcing to the glacier model (to be discussed later).
OK.

Line 168: Comment on uncertainties associated with estimation of topographic parameters using this method.
I have added the sentence: *Since the interest is in the bulk mass budget of a basin, the uncertainties in the geometric parameters have a limited effect only*

Line 174: delete "The formulation of a calving law has been a controversial issue since a long time."
OK.

Line 194: It should be noted that comprehensive numerical models have not yet been used to validate...
To validate what ? The model or the fluctuations ?  I think my formulation is ok.

Line 203: Should this title read "main stream"
Yes, of course; thanks !.

Line 209: "is not met"
OK.

Line 243: "is no exception"
OK.

Line 245: Compared to the glacier length,
OK.

Line 271: respectively
OK.

Line 303: there has been a long-term trend
OK.

Line 304: response time scale
OK.

Line 304: Where do you get this time scale from?
Admittedly a guess based on the size and slope of the glacier. However, I don't think a more precise calculation, if possible, is useful here.

Line 321: anomalies
OK.

Line 335: successful
OK.

Line 373: delete apparently
OK.

Line 377: loss
I am not sure... I think the word 'net' is useful here ! ?

Line 431: projections of future climate change scenarios
OK.

Line 469-470: Why is this comparison interesting?
I am surprised by this comment. Of course this comparison is interesting, because we want to know how important surging is for the long-term evolution of a glacier and how this differs from glacier to glacier.

Line 502: observed
OK.

[revised manuscript text omitted]

---

## Author Response (AR3)

Monacobreen is a 40 km long surge-type tidewater glacier in northern Spitsbergen. During 1991-1997 Monacobreen surged and advanced by about 2 km, but the front did not reach the maximum Little Ice Age (LIA) stand. Since 1997 the glacier front is retreating at a fast rate (~ 125 m/a). The questions addressed in this study are: (1) Can the late Holocene behaviour of Monacobreen be understood in terms of climatic forcing ?, and (2) What will be the likely evolution of this glacier for different scenarios of future climate change ?

Monacobreen is modelled with a Minimal Glacier Model, including a parameterization of the calving process as well as the effect of surges. The model is driven by an Equilibrium Line Altitude (ELA) history derived from lake sediments of a nearby glacier catchment, in combination with meteorological data from 1899 onwards. The simulated glacier length is in good agreement with the observations: the maximum LIA stand, the front position at the end of the surge, and the 2.5 km retreat after the surge (1997-2016) are well reproduced (the mean difference between observed and simulated glacier length being 6% when scaled with the total retreat during 1900-2016). The effect of surging is limited. Directly after a surge the initiated mass-balance pertubation due to a lower mean surface elevation is about $-0.13$ m w.e. $a^{-1}$, which only has a small effect on the long-term evolution of the glacier. The simulation suggests that the major growth of Monacobreen after the Holocene Climatic Optimum started around 1500 BCE. Monacobreen became a tidewater glacier around 500 BCE, and reached a size comparable to the present state around 500 CE. For the mid-B2 scenario (IPCC, 2013), which corresponds to a ~2 m $a^{-1}$ rise of the ELA, the model predicts a volume loss of 20 to 30 % by the year 2100 (relative to the 2017 volume). For a ~4 m $a^{-1}$ rise in the ELA this is 30 to 40 %. However, much of the response to 21st century warming will still come after 2100.